# Reward Machines for Deep RL
# in Noisy and Uncertain Environments

**Andrew C. Li**
University of Toronto
Vector Institute

**Zizhao Chen**[*]
Cornell University

**Toryn Q. Klassen**[†]
University of Toronto
Vector Institute

**Pashootan Vaezipoor**
Georgian.io
Vector Institute

**Rodrigo Toro Icarte**
Pontificia Universidad Católica de Chile
Centro Nacional de Inteligencia Artificial

**Sheila A. McIlraith**[†]
University of Toronto
Vector Institute

## Abstract

Reward Machines provide an automaton-inspired structure for specifying instructions, safety constraints, and other temporally extended reward-worthy behaviour. By exposing the underlying structure of a reward function, they enable the decomposition of an RL task, leading to impressive gains in sample efficiency. Although Reward Machines and similar formal specifications have a rich history of application towards sequential decision-making problems, prior frameworks have traditionally ignored ambiguity and uncertainty when interpreting the domain-specific vocabulary forming the building blocks of the reward function. Such uncertainty critically arises in many real-world settings due to factors like partial observability or noisy sensors. In this work, we explore the use of Reward Machines for Deep RL in noisy and uncertain environments. We characterize this problem as a POMDP and propose a suite of RL algorithms that exploit task structure under uncertain interpretation of the domain-specific vocabulary. Through theory and experiments, we expose pitfalls in naive approaches to this problem while simultaneously demonstrating how task structure can be successfully leveraged under noisy interpretations of the vocabulary.

**Code and videos are available at** `https://github.com/andrewli77/reward-machines-noisy-environments`.

## 1 Introduction

Formal languages, including programming languages such as Python and C, have long been used for objective specification in sequential decision making. Using a vocabulary of domain-specific properties, expressed as propositional variables, formal languages like Linear Temporal Logic (LTL) [44] capture complex temporal patterns—such as the objectives of an agent—by composing variables via temporal operators and logical connectives. These languages provide well-defined semantics while enabling semantics-preserving transformations to normal-form representations such as automata, which can expose the discrete structure underlying an objective to a decision-making agent. One such representation is the increasingly popular Reward Machine, which combines expression of rich temporally extended (non-Markovian) objectives via automata with algorithmic techniques such as automated reward shaping, task decomposition, and counterfactual learning updates to garner

---

[*]Work done while at the University of Toronto.

[†]Also affiliated with the Schwartz Reisman Institute for Technology and Society.
  Correspondence to `andrewli@cs.toronto.edu`.

significant improvements in sample efficiency (e.g., [52, 54, 18]). Importantly, Reward Machines can be specified directly, constructed via translation from any regular language (including variants of LTL), synthesized from high-level planning specifications, or learned from data [7]. For these reasons, formal languages, and in particular, Reward Machines, have been adopted across a diversity of domains, ranging from motion planning [30, 15, 50] and robotic manipulation [8, 26, 31] to, more recently, general deep Reinforcement Learning (RL) problems [e.g., 1, 34, 35, 37, 52, 53, 21, 64, 29, 61, 32, 22, 62, 63, 17, 46, 11, 28, 41, 49, 59, 12, 66, 8, 10, 38, 18, 48, 65, 58, 55].

Importantly, formal language frameworks for deep RL require an interpretation of the domain-specific vocabulary grounded in the RL environment. This is captured by a *labelling function*, a mapping from environment states to the truth or falsity of abstract propositions that constitute the building blocks of objective specifications. However, practical real-world environments are often partially observable and rely on high-dimensional sensor data such as images. As a result, labelling functions are, by necessity, noisy and uncertain, compromising the application of formal languages such as Reward Machines or LTL for objective specification. Consider an autonomous vehicle, whose desired behaviour at an intersection can be formally specified using temporal logic [5, 39]. In the real world, key determinations—whether a pedestrian is crossing, the colour of the light, the intent of other vehicles, and so on—must be made based on noisy or obstructed LiDAR and camera sensors, and may therefore be noisy or uncertain themselves. To address this problem, while benefiting from the advantages of Reward Machines, we make the following contributions.

**(1)** We propose a deep RL framework for Reward Machines in settings where the evaluation of domain-specific vocabulary is uncertain, characterizing the RL problem in terms of a POMDP. To our knowledge, this is the first deep RL framework for Reward Machines that broadly supports the imperfect detection of propositions, allowing us to extend Reward Machines to general partially observable environments.

**(2)** We propose and analyze a suite of RL algorithms that exploit Reward Machine structure under noisy and uncertain interpretations of the vocabulary. We show how preexisting *abstraction models*—noisy estimators of abstract, task-relevant features that may manifest as pretrained neural networks, sensors, heuristics, or otherwise—can be brought to bear to improve learning efficiency.

**(3)** We theoretically and experimentally evaluate our proposed RL algorithms. Theoretically, we discover a pitfall of naively leveraging standard abstraction models—namely, that errors from repeated queries of a model are correlated rather than i.i.d. We show that this can have serious ramifications, including unintended or dangerous outcomes, and demonstrate how this issue can be mitigated. Experimentally, we consider a variety of challenging domains involving partial observability and high-dimensional observations. Results show that our algorithms successfully leverage task structure to improve sample efficiency and total reward under uncertain interpretations of the vocabulary.

## 2 Background

**Notation.** Given a set of random variables, $X$, $\Delta X$ is the set of distributions over $X$; $\tilde{(\cdot)}$ denotes a particular distribution; and for a categorical distribution $\tilde{w} \in \Delta X$ and some $x \in X$, we denote $\tilde{w}[x]$ as the probability of $x$ under $\tilde{w}$. We use $x_{i:j}$ as a shorthand for the sequence $x_i, \ldots, x_j$.

**POMDPs.** A *Partially Observable Markov Decision Process* (POMDP) $\langle S, O, A, P, R, \omega, \mu \rangle$ [3] is defined by the state space $S$, observation space $O$, action space $A$, reward function $R$, transition distribution $P : S \times A \rightarrow \Delta S$, observation distribution $\omega : S \rightarrow \Delta O$, and initial state distribution $\mu \in \Delta S$. An *episode* begins with an initial state $s_1 \sim \mu(\cdot)$ and at each timestep $t \geq 1$, the agent observes an observation $o_t \sim \omega(s_t)$, performs an action $a_t \in A$, transitions to the next state $s_{t+1} \sim P(s_t, a_t)$, and receives reward $r_t = R(s_t, a_t, s_{t+1})$. Denote the agent's observation-action history at time $t$ by $h_t = (o_1, a_1, \ldots, o_{t-1}, a_{t-1}, o_t)$ and the set of all possible histories as $H$. A fully observable *Markov Decision Process* (MDP) serves as an important special case where the observation at each time $t$ is $o_t = s_t$.

**Reward Machines.** A *Reward Machine* (RM) [52] is a formal automaton representation of a non-Markovian reward function that captures temporally extended behaviours. Formally, an RM $\mathcal{R} = \langle U, u_1, F, \mathcal{AP}, \delta_u, \delta_r \rangle$, where $U$ is a finite set of states, $u_1 \in U$ is the initial state, $F$ is a finite set of terminal states (disjoint from $U$), $\mathcal{AP}$ is a finite set of atomic propositions representing the occurrence of salient events in the environment, $\delta_u : U \times 2^{\mathcal{AP}} \rightarrow (U \cup F)$ is the state-transition

function, and $\delta_r : U \times 2^{\mathcal{AP}} \to \mathbb{R}$ is the state-reward function. Each transition in an RM is labelled with a scalar reward along with a propositional logic formula over $\mathcal{AP}$, while accepting states represent task termination.

**Labelling Function.** While an RM captures the high-level structure of a non-Markovian reward function, concrete rewards in an environment are determined with the help of a *labelling function* $\mathcal{L} : S \times A \times S \to 2^{\mathcal{AP}}$, a mapping that abstracts state transitions $(s_{t-1}, a_{t-1}, s_t)$ in the environment to the subset of propositions that hold for that transition. To obtain rewards, the sequence of environment states and actions $s_1, a_1, \ldots, s_t, a_t, s_{t+1}$ are labelled with propositional evaluations $\sigma_{1:t}$, where $\sigma_i = \mathcal{L}(s_i, a_i, s_{i+1}) \in 2^{\mathcal{AP}}$. A sequence of transitions in the RM are then followed based on $\sigma_{1:t}$ to produce the reward sequence $r_{1:t}$. In an MDP environment, rewards specified by an RM are Markovian over an extended state space $S \times U$; hence, there is an optimal policy of the form $\pi(a_t|s_t, u_t)$ where $u_t \in U$ is the RM state at time $t$ [52]. During execution, $u_t$ can be recursively updated given $u_{t-1}$ by querying $\mathcal{L}$ after each environment transition.

# 3 Problem Framework

## 3.1 Formalization

We formalize the problem of solving an RM task under an uncertain interpretation of the vocabulary. Consider an agent acting in a POMDP environment (without the reward function) $\mathcal{E} = \langle S, O, A, P, \omega, \mu \rangle$. We define rewards $r_t$ based on an RM $\mathcal{R} = \langle U, u_1, F, \mathcal{AP}, \delta_u, \delta_r \rangle$ interpreted under a ground-truth labelling function $\mathcal{L} : S \times A \times S \to 2^{\mathcal{AP}}$ as described in Section 2.

An important aspect of our framework is that $\mathcal{L}$ is not made accessible to the agent. Instead, we make the weaker assumption that the agent can query an *abstraction model* $\mathcal{M} : H \to Z$. Here, $\mathcal{M}$ captures the agent's preexisting knowledge over how a set of high-level features $Z$ are grounded within the environment. Abstraction models are easier to obtain than ground-truth labelling functions for several reasons: they take

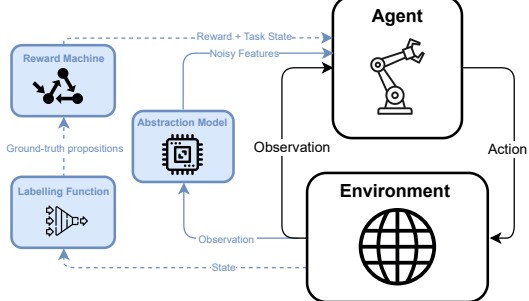

Figure 1: The *Noisy Reward Machine Environment* framework. Blue elements highlight differences with respect to a standard RL framework. Dashed lines ( - - - - ) indicate that an element is required during training but not deployment.

as input observable histories $H$ rather than states $S$, they can map to any feature space $Z$, and crucially, we allow their outcomes to be incorrect or uncertain. Note that the definition of abstraction models is quite general—in the real world, they might manifest as pretrained foundation models [45, 36, 4, 14], sensors [16], task-specific classifiers [20], or so on.

We refer to the tuple $\mathcal{P} = \langle \mathcal{E}, \mathcal{R}, \mathcal{L}, \mathcal{M} \rangle$ as a *Noisy Reward Machine Environment* (depicted in Figure 1). Given $\mathcal{P}$, our goal is to obtain a policy $\pi(a_t|h_t, z_{1:t})$ based on observations and outputs from the abstraction model up to time $t$ that maximizes the expected discounted return $\mathbb{E}_\pi[\sum_{t=0}^{\infty} \gamma^t r_t]$ for some discount factor $\gamma \in (0, 1]$. In this work, we assume $\pi$ is trained via RL, and we assume that the ground-truth rewards $r_i$ are observable during training only. Notably, the trained policy $\pi$ can be deployed without access to the ground-truth labelling function $\mathcal{L}$.

## 3.2 Running Example

The *Gold Mining Problem* (Figure 2) serves as a running example of a Noisy RM Environment. A mining robot operates in a grid with a non-Markovian goal: dig up at least one chunk of gold ( 🪙 ) and deliver it to the depot ( 🏠 ). The environment is an MDP where the robot observes its current grid position and its actions include moving in the cardinal directions and digging. A labelling function $\mathcal{L}$ associates the propositions $\mathcal{AP} = \{$ 🪙 , 🏠 $\}$ with grid states and actions as follows: 🪙 holds when the robot digs in the rightmost row, and 🏠 holds when the robot is at the bottom-left cell.

However, the robot does not have access to $\mathcal{L}$ and cannot reliably distinguish gold from iron pyrite. Thus, it cannot ascertain whether it has obtained gold during the course of an episode. Luckily, the robot can make an educated guess as to whether a cell contains gold, which is captured by an

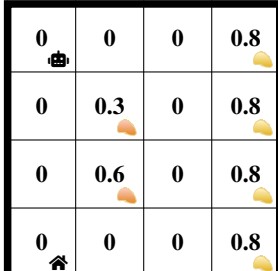 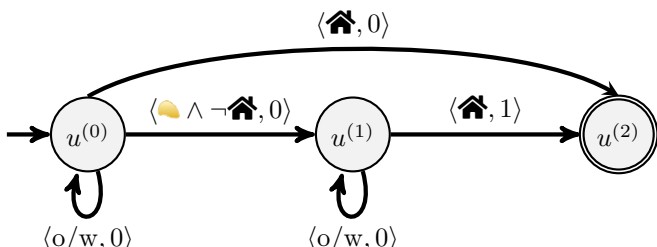

Figure 2: The *Gold Mining Problem* is a Noisy RM Environment where the agent's interpretation of the vocabulary is uncertain. **Left:** The four rightmost cells yield gold ( 🪙 ) while two cells in the second column yield iron pyrite, which has no value. The agent cannot reliably distinguish between the two metals—cells are labelled with the probability the agent *believes* it yields gold. **Right:** The RM emits a (non-Markovian) reward of 1 for collecting gold and delivering it to the depot (🏠).

abstraction model $\mathcal{M} : H \to [0, 1]$ mapping the robot's current position (while ignoring the rest of the history) to its belief that the cell contains gold.

Note that if the agent *could* observe $\mathcal{L}$, then it could learn an optimal Markovian policy $\pi(a_t|s_t, u_t)$ with a relatively simple form ($u_t$ is easily computed with access to $\mathcal{L}$). Intuitively, such a policy should collect gold while in RM state $u^{(0)}$ and head to the depot while in RM state $u^{(1)}$. Unfortunately, when the agent does not have access to $\mathcal{L}$, we cannot directly learn a policy with the simpler Markovian form above. In the following sections, we show how the agent's noisy belief captured by the abstraction model $\mathcal{M}$ can be leveraged to simplify the learning problem.

## 4 Noisy RM Environments as POMDPs

We start with an analysis of the Noisy RM Environment framework, contrasting it with a standard RM framework. We ask: **(1)** What is the optimal behaviour in a Noisy RM Environment? **(2)** How does the abstraction model $\mathcal{M}$ affect the problem? **(3)** How does not observing the ground-truth labelling function $\mathcal{L}$ affect the problem? We provide proofs for all theorems in Appendix A.

Observe that uncertainty in the propositional values is only relevant insofar as it influences the agent's belief about the current RM state $u_t$ since rewards from an RM $\mathcal{R}$ are Markovian over extended states $(s_t, u_t) \in S \times U$. Our first result is that a Noisy RM Environment $\langle \mathcal{E}, \mathcal{R}, \mathcal{L}, \mathcal{M} \rangle$ can be reformulated into an equivalent POMDP with state space $S \times U$ and observation space $O$ (Theorem 4.1). Here, we say two problems are equivalent if there is a bijection between policies for either problem such that the policies have equal expected discounted return and behave identically given the same history $h_t$. Thus, optimal behaviour in a Noisy RM Environment can be reduced to solving a POMDP.

**Theorem 4.1** A Noisy RM Environment $\langle \mathcal{E}, \mathcal{R}, \mathcal{L}, \mathcal{M} \rangle$ is equivalent to a POMDP over state space $S \times U$ and observation space $O$.

One may notice that the abstraction model $\mathcal{M}$ doesn't appear in the POMDP reformulation at all. We later show that an appropriate choice of $\mathcal{M}$ can improve policy learning in practice, but this choice ultimately does not change the optimal behaviour of the agent (Theorem 4.2).

**Theorem 4.2** (*Does the choice of $\mathcal{M}$ affect optimal behaviour?*) Let $\mathcal{P}$ be a Noisy RM Environment $\langle \mathcal{E}, \mathcal{R}, \mathcal{L}, \mathcal{M} \rangle$, and $\mathcal{P}'$ be a Noisy RM Environment $\langle \mathcal{E}, \mathcal{R}, \mathcal{L}, \mathcal{M}' \rangle$. Then $\mathcal{P}$ and $\mathcal{P}'$ are equivalent.

We also contrast our proposed framework, where the agent does not have direct access to $\mathcal{L}$, with prior RM frameworks where the agent does. We show that this difference does not affect the optimal behaviour in MDP environments, but can affect the optimal behaviour in POMDPs (Theorem 4.3).

**Theorem 4.3** (*Does observing $\mathcal{L}$ affect optimal behaviour?*) Let $\mathcal{P}$ be a Noisy RM Environment $\langle \mathcal{E}, \mathcal{R}, \mathcal{L}, \mathcal{M} \rangle$. Consider a problem $\mathcal{P}'$ that is identical to $\mathcal{P}$ except that the agent at time $t$ additionally observes $\mathcal{L}(s_t, a_t, s_{t+1})$ after taking action $a_t$ in state $s_t$. If $\mathcal{E}$ is an MDP, then $\mathcal{P}$ and $\mathcal{P}'$ are equivalent. If $\mathcal{E}$ is a POMDP, $\mathcal{P}$ and $\mathcal{P}'$ may be non-equivalent.

# 5 Method

In this section, we consider how to train policies that do not require the ground-truth labelling function $\mathcal{L}$ to act. Given Theorem 4.1, in principle we can apply any deep RL approach suitable for POMDPs, such as a recurrent policy that conditions on past actions and observations. However, such a learning algorithm may be inefficient as it doesn't exploit the known RM structure at all. Motivated by the observation that the RM state $u_t$ is critical for decision making, we instead propose a class of policies that decouple inference of the RM state $u_t$ (treated as an unknown random variable) and decision making into two separate modules (Algorithm 1).

**Input:** Abstraction model $\mathcal{M} : H \to Z$, Inference module $f : Z^+ \to \Delta U$.

Initialize policy $\pi : H \times \Delta U \to \Delta A$.
**for** each episode **do**
  Observe $o_1$.
  $h_1 \leftarrow (o_1)$
  **for** each time $t = 1, 2, \ldots$ **do**
    $z_t \leftarrow \mathcal{M}(h_t)$
    $\tilde{u}_t \leftarrow f(z_{1:t})$
    Execute action $a_t \sim \pi(\cdot \mid h_t, \tilde{u}_t)$.
    Get reward $r_t$ and observation $o_{t+1}$.
    Update $\pi$ using RL.
    $h_{t+1} \leftarrow h_t + (a_t, o_{t+1})$

Algorithm 1: On-policy RL that decouples RM state inference using an abstraction model $\mathcal{M}$ and decision making.

The *inference* module models a belief $\tilde{u}_t \in \Delta U$ of the current RM state $u_t$ over the course of an episode with the help of an abstraction model $\mathcal{M}$. More precisely, the inference module is a function $f : Z^+ \to \Delta U$ mapping the history of outputs from the abstraction model to a belief over RM states. The inference objective is to recover the (policy-independent) distribution $\Pr(u_t | h_t)$, marginalized over all possible state trajectories $\tau_t = (s_1, a_1, \ldots, s_{t-1}, a_{t-1}, s_t)$:

$$\Pr(u_t | h_t) = \int \Pr(u_t | \tau_t) p(\tau_t | h_t) d\tau_t$$

Here $\Pr(u_t | \tau_t)$ is deterministic—it is the RM state given the trajectory (under the ground-truth labelling function)—while the probability density function $p(\tau_t | h_t)$ depends on the POMDP transition function and observation probabilities. The *decision-making* module is a policy $\pi(a_t | h_t, \tilde{u}_t)$ that then leverages the inferred belief $\tilde{u}_t$. Below, we describe three inference modules that leverage different forms of abstraction models $\mathcal{M} : H \to Z$ to predict $\tilde{u}_t$.

## 5.1 Naive

Suppose that in lieu of the ground-truth labelling function $\mathcal{L}$, we have a noisy estimator of $\mathcal{L}$ that predicts propositional evaluations based on the observation history. This is captured via an abstraction model of the form $\mathcal{M} : H \to 2^{\mathcal{AP}}$ that makes discrete (and potentially incorrect) predictions about the propositions $\mathcal{L}(s_{t-1}, a_{t-1}, s_t)$ that hold at time $t$, given the history $h_t$. Then, we can recursively model a discrete prediction of the RM state $\hat{u}_t \in U$ (which can be seen as a belief over $U$ with full probability mass on $\hat{u}_t$) using outputs of $\mathcal{M}$ in place of $\mathcal{L}$.

**Method 1** (*Naive*) Given $\mathcal{M} : H \to 2^{\mathcal{AP}}$, predict a discrete RM state $\hat{u}_t \in U$ as follows. Set $\hat{u}_1 = u_1$. For $t > 1$, predict $\hat{u}_t = \delta_u(\hat{u}_{t-1}, \mathcal{M}(h_t))$.

A weakness of this approach is that it does not represent the uncertainty in its prediction. Furthermore, since RM state predictions are updated recursively, an error when predicting $\hat{u}_t$ will propagate to all future predictions $(\hat{u}_{t+1}, \hat{u}_{t+2}, \ldots)$.

**Example 5.1** Returning to the running example, suppose the agent uses its belief of locations yielding gold to derive an abstraction model $\mathcal{M} : H \to 2^{\mathcal{AP}}$. For a history $h_t$, $\mathcal{M}$ takes the current grid position $s_t$ and predicts 🟡 is true if it believes there is at least a 50% chance it yields gold when it performs a digging action. We assume the agent can always predict 🏠 correctly. Observing Figure 2, we see that $\mathcal{M}$ agrees with $\mathcal{L}$ in all cases except at one cell where there is actually iron pyrite (that the agent believes has gold with 0.6 probability). Given a trajectory where the agent mines at this cell, Naive would erroneously assume gold was obtained.

## 5.2 Independent Belief Updating (IBU)

In order to capture the uncertainty in the belief over the RM state $u_t$, one may instead wish to model a probability distribution $\tilde{u}_t \in \Delta U$. Given an abstraction model of the form $\mathcal{M} : H \to \Delta(2^{\mathcal{AP}})$ that

Table 1: Comparison of inference modules. For each, we highlight its prerequisite abstraction model, the target feature the abstraction model aims to predict, and its *consistency* in MDPs and POMDPs.

| INFERENCE MODULE | ABSTRACTION MODEL | TARGET | CONSISTENT (MDPs) | CONSISTENT (POMDPs) |
|---|---|---|---|---|
| NAIVE | $H \to 2^{\mathcal{AP}}$ | $\mathcal{L}(s_{t-1}, a_{t-1}, s_t)$ | ✓ | ✗ |
| IBU | $H \to \Delta(2^{\mathcal{AP}})$ | $\mathcal{L}(s_{t-1}, a_{t-1}, s_t)$ | ✓ | ✗ |
| TDM | $H \to \Delta U$ | $u_t$ | ✓ | ✓ |

predicts probabilities over possible propositional evaluations $\mathcal{L}(s_{t-1}, a_{t-1}, s_t)$, an enticing approach is to derive $\tilde{u}_t$ by probabilistically weighing all possible RM states at time $t-1$ according to the previous belief $\tilde{u}_{t-1}$ along with all possible evaluations of $\mathcal{L}(s_{t-1}, a_{t-1}, s_t)$ according to $\mathcal{M}$.

**Method 2** (*IBU*) Given $\mathcal{M} : H \to \Delta(2^{\mathcal{AP}})$, predict a distribution over RM states $\tilde{u}_t \in \Delta U$ as follows. Set $\tilde{u}_1[u_1] = 1$ and $\tilde{u}_1[u] = 0$ for $u \in U \setminus \{u_1\}$. For $t > 1$, set

$$\tilde{u}_t[u] = \sum_{\sigma \in 2^{\mathcal{AP}}, u' \in U} \mathbb{1}[\delta_u(u', \sigma) = u] \cdot \tilde{u}_{t-1}[u'] \cdot \mathcal{M}(h_t)[\sigma]$$

On the surface, IBU may appear to solve the error propagation issue of the Naive approach by softening a discrete belief into a probabilistic one. Surprisingly, updating beliefs $\tilde{u}_t$ in this manner can still result in a belief that quickly diverges from reality with increasing $t$. This is because IBU fails to consider that propositional evaluations are linked, rather than independent. Since the computation of $\tilde{u}_t$ aggregates $t$ queries to $\mathcal{M}$, noise in the outputs of $\mathcal{M}$ can dramatically amplify if the *correlation* between these noisy outputs is not considered. This is best illustrated by an example.

**Example 5.2** The mining agent now considers a probability distribution over propositional assignments of $\{🪙, 🏠\}$. We assume the agent always perfectly determines 🏠 and applies its belief of locations yielding gold; e.g., digging in the cell the agent believes has gold with 0.3 probability yields the distribution $(\emptyset : 0.7, \{🪙\} : 0.3, \{🏠\} : 0, \{🪙, 🏠\} : 0)$. Consider a trajectory where the agent digs at this cell multiple times. After mining once, IBU updates the RM state belief $\tilde{u}_t$ to reflect a 0.3 probability of having obtained gold. After mining twice, this increases to 0.51 and in general, the belief reflects a $1 - 0.7^k$ probability after mining $k$ times. In reality, mining more than once at this square should not increase the probability beyond 0.3 since all evaluations of 🪙 at that cell are linked—they are all true, or they are all false.

### 5.3 Temporal Dependency Modelling (TDM)

Example 5.2 demonstrates a challenge when aggregating multiple predictions from a noisy classifier— the noise may be correlated between evaluations. One solution in the Gold Mining Problem is to update the RM state belief only the first time the agent digs at any particular square, accounting for the fact that all evaluations of 🪙 in that state yield the same result. Indeed, this type of solution is used by many approaches in tabular MDPs with noisy propositional evaluations [19, 60], but this solution does not scale to infinite state spaces where propositional evaluations may be arbitrarily linked between "similar" pairs of states.

We instead consider an inference module that uses an abstraction model $\mathcal{M} : H \to \Delta U$ designed to directly predict a distribution over RM states given the history. Such an abstraction model might manifest as a meta-classifier that aggregates outputs from another model but corrects for the correlation in these outputs. Another way to obtain $\mathcal{M}$ is to train a recurrent neural network [27] end-to-end given a dataset of histories $h_t$ and their associated (ground-truth) RM states $u_t$. Given an abstraction model $\mathcal{M} : H \to \Delta U$, TDM simply returns the output of $\mathcal{M}$.

**Method 3** (*TDM*) Given an abstraction model of the form $\mathcal{M} : H \to \Delta U$ predict $\mathcal{M}(h_t)$ directly.

### 5.4 Comparison of Inference Modules

At first glance, it may seem challenging to compare different inference modules since they may operate under different abstraction models. Our goal is to elucidate the relatives advantages of each approach to better inform its use. We begin by considering a theoretical property of inference modules,

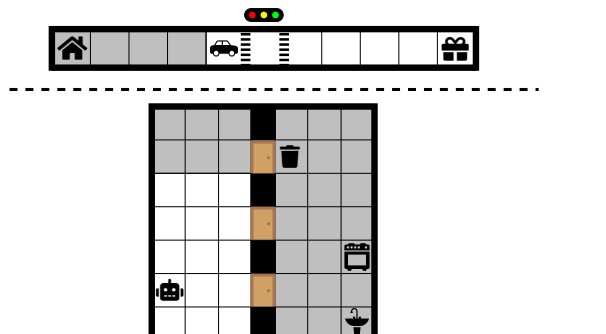
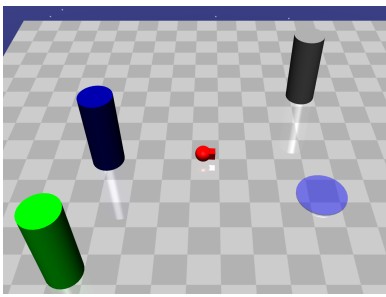

Figure 3: Traffic Light (top left) and Kitchen (bottom left), are MiniGrids with image observations, where key propositions are partially observable. Colour Matching (right) is a MuJoCo robotics environment where the agent must identify colour names by their RGB values to solve a task.

with our conclusions summarized in Table 1. We consider how accurately an inference module models its target distribution $\Pr(u_t|h_t)$. This largely depends on the veracity of the abstraction model $\mathcal{M}$, but nonetheless, it is desirable that the inference module precisely recovers $\Pr(u_t|h_t)$ under an ideal abstraction model $\mathcal{M}^*$. If this is possible, then we say that an inference module is *consistent*.

**Definition 5.3** (*Consistency*) Consider an inference module $f : Z^+ \to \Delta U$. $f$ is consistent if there exists some $\mathcal{M}^* : H \to Z$ such that for every history $h_t \in H$, running $f$ on $h_t$ using $\mathcal{M}^*$ as the abstraction model results in the belief $\Pr(u_t|h_t)$.

In the case that $\mathcal{E}$ is an MDP, Naive, IBU, and TDM are all consistent since there exists an abstraction model that can recover the ground-truth labelling function $\mathcal{L}$ (for Naive and IBU) and $u_t$ (for TDM) with certainty. However, in the general case when $\mathcal{E}$ is a POMDP, only TDM remains consistent. Proofs and counterexamples are provided in Appendix A.

# 6 Experiments

Our experiments assess the approaches in Section 5 in terms of RL sample efficiency and final total return, as well as accuracy in predicting a belief over RM states. We examine whether these methods are robust to uncertainty in the outputs of abstraction models, and whether they offer advantages over end-to-end RL algorithms that do not attempt to exploit task structure. Our experimental settings involve challenges that arise when scaling to complex, real-world environments: temporally extended reasoning, partial observability, high-dimensional observations, and sparse rewards.

## 6.1 Environments

Our environments include the *Gold Mining* Problem as a toy environment, along with two MiniGrid [9] environments with image observations and a MuJoCo robotics environment (Figure 3). Full details on the environments are provided in Appendix B.

*Traffic Light* is a partially observable MiniGrid where the agent must drive along a road to pick up a package and then return home. A traffic light along the road cycles between green, yellow, and red at stochastic intervals and the agent receives a delayed penalty if it runs the red light. The agent only has a forward-facing camera and it can drive forwards, backwards, wait, or make a U-turn. We encode this task with an RM (Figure 7) with propositions for running a red light, collecting the package, and returning home. Crucially, the agent does not observe the light colour when entering the intersection in reverse, causing the agent to be uncertain about the evaluation of the red light proposition.

*Kitchen* is a partially observable MiniGrid where a cleaning agent starts in the foyer of a home and is tasked with cleaning the kitchen before leaving. There are three chores: the agent must make sure the dishes are washed, the stove is wiped, and the trash is emptied, but not every chore necessarily requires action from the agent (e.g. there might be no dirty dishes to begin with). However, the agent doesn't know how clean the kitchen is until it enters it. For each chore, a proposition represents whether that chore is "done" (e.g. the dishes are clean) in the current state of the environment—thus, the propositional evaluations are unknown to the agent until it enters the kitchen. The RM for this task (omitted due to its size) uses the automaton state to encode the subset of chores that are currently done and gives a reward of 1 for leaving the house once the kitchen is clean.

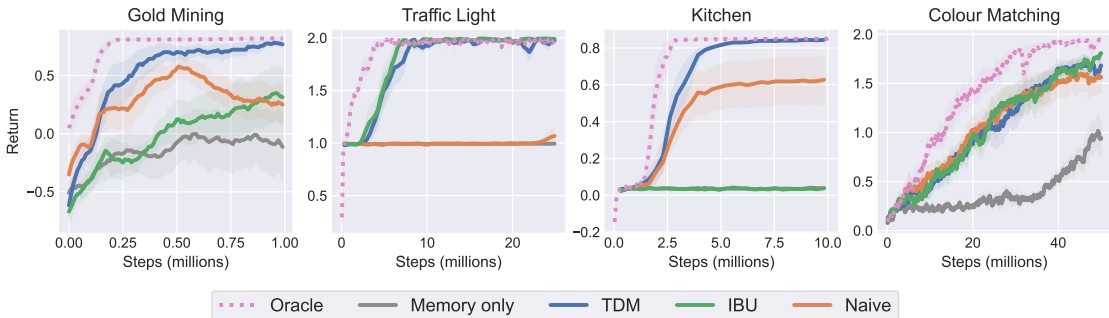

Figure 4: RL curves averaged over 8 runs (shaded regions show standard error). TDM performs well in all domains, in the absence of the ground-truth labelling function, while Recurrent PPO fails.

In *Colour Matching*, the agent controls a robot car while observing LiDAR and RGB observations of nearby objects. The agent observes an instruction with the name of a colour (e.g. "blue") and it must touch only that pillar and then enter the portal. Propositions for each pillar evaluate whether the agent is touching it. The colours of the pillars (and in the instruction) are randomized from a set of 18 possible colours in each episode, so to reliably solve the task, the agent must learn the correct associations of colour names to their RGB values.

## 6.2 Baselines

We consider the methods *Naive*, *IBU*, and *TDM* from Section 5 that use the abstraction models described below. For RL experiments, we baseline against a *Memory-only* method for general POMDPs that does not exploit the RM structure. As an upper bound on performance, we compare against an *Oracle* agent that has access to the ground-truth labelling function. In the toy Gold Mining Problem, policies are trained using Q-learning [51] with linear function approximation [40]. In all other domains, policies are neural networks trained with PPO [47], and the Memory-only policy is Recurrent PPO [25], a popular state-of-the-art deep RL method for general POMDPs.

## 6.3 Abstraction Models

In the Gold Mining Problem, we consider toy abstraction models based on the probabilities in Figure 2 as described in the running examples. TDM is equivalent to IBU except it only updates the RM state belief when digging for the first time at the current cell.

In all other domains, abstractions models are neural networks trained via supervised learning. We collect training datasets comprising 2K episodes in each domain (equalling 103K interactions in Traffic Light, 397K interactions in Kitchen, and 3.7M interactions in Colour Matching), along with validation and test datasets of 100 episodes each. This data is generated by a random policy and labelled with propositional evaluations from $\mathcal{L}$ and ground-truth RM states. To obtain abstraction models, we train classifiers on their respective target labels and we select optimal hyperparameters according to a grid search.[1] We note that all abstraction models are trained on equivalent data, ensuring a fair comparison between different inference modules. To verify that abstraction models are indeed noisy, we use the test set to evaluate the precision and recall of a classifier trained to predict propositional evaluations (Figure 6). We find that key propositions towards decision making, such as running a red light in Traffic Light, or whether the chores are done in Kitchen, are uncertain.

## 6.4 Results: Reinforcement Learning

We report RL learning curves for each method and environment in Figure 4. The key results are:

**(R1) TDM performs well in all domains.** Using only a noisy abstraction model, TDM achieves similar sample efficiency and final performance to the Oracle agent that has access to the ground truth.

---

[1] For RL training on Traffic Light and Kitchen, we continue to finetune the abstraction models using data collected by the policy— this is to verify that partial observability is difficult to handle even with virtually unlimited data and no distributional shift.

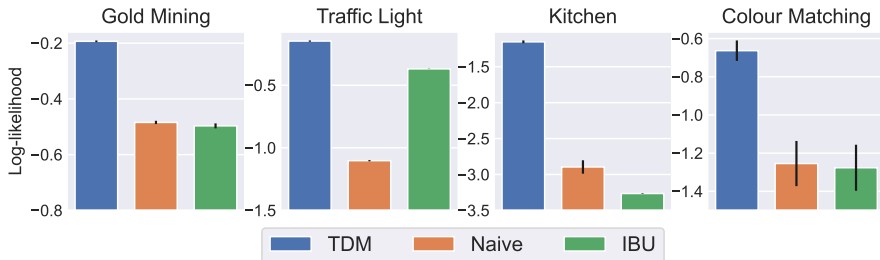

Figure 5: Accuracy of inference modules measured by log-likelihood (higher is better) of the true RM state, averaged over 8 runs with lines showing standard error. TDM predicts the RM state belief more accurately than Naive and IBU.

**(R2): RL makes little progress on any domain when the task structure is not exploited.** Specifically, Memory only (recurrent PPO in the deep RL environments) fails since it does not leverage the temporal and logical structure afforded by the RM.

**(R3): The performance of Naive and IBU depends on the specific environment.** When the simplifying assumptions made by Naive or IBU are reasonable, these approaches can perform well. However, the use of these approaches under noisy abstraction models can also lead to dangerous or unintended outcomes (see Appendix B.5 for a discussion).

We now highlight the most notable qualitative behaviours that were observed. For a more in-depth discussion, refer to Appendix B.5. In Gold Mining, Naive often digs at the nearby cell believed to yield gold with 0.6 probability (but in actuality yielding iron pyrite) before immediately heading to the depot, and IBU repeatedly digs at the same cell to increase its belief of having obtained gold. On the other hand, TDM adopts a robust strategy of mining at multiple different cells to maximize its chance of having obtained gold. In Traffic Light, Naive often runs the red light by driving in reverse to get through the intersection faster. This shortcoming stems from its inability to represent uncertainty about running the red light. In Kitchen, IBU often stays in the foyer without ever entering the kitchen. Despite this, the RM state belief erroneously reflects that all chores have a high probability of being done. This is similar to Example 5.2—each chore is initially "clean" with some small probability, and this is compounded over time by the incorrect independence assumption. In reality, the state of the chore is linked between timesteps (and doesn't change unless the agent intervenes).

## 6.5   Results: RM State Belief Inference

We compare the inference modules Naive, IBU, and TDM in terms of their predictive accuracy of the RM state (Figure 5). We evaluate each approach on a test set of 100 fixed trajectories generated by a random policy (following the same distribution as the data used to train the abstraction models). Since each inference module aims to capture a belief, we evaluate them according to the *log-likelihood* of the true RM state under the belief, averaging over the predictions at all timesteps. To avoid $\log 0$ when Naive makes an incorrect prediction, we lower bound all log-likelihoods at $\ln 0.01$.

**(R4): TDM is more accurate when predicting a belief over RM states compared to Naive or IBU on all domains.**

## 6.6   Results: Vision-Language Models as Zero-Shot Abstraction Models

As an additional experiment, we assess whether GPT-4o can serve as an effective zero-shot abstraction model in the Traffic Light domain. We render RGB images of the environment and use GPT-4o to evaluate propositions described through text. With this abstraction model, we evaluate Naive and IBU on the test set from Section 6.5. We baseline these against the inference modules from Section 6.5 that use abstraction models trained via supervised learning, as well as abstraction models with randomly initialized weights. Results and further details are provided in Figure 8 of the Appendix.

**(R5): GPT-4o is an effective zero-shot abstraction model for Naive and IBU.** As an abstraction model, GPT-4o is nearly as effective as a custom model trained from ground-truth data, and is significantly more effective than a randomly initialized neural network.

# 7 Related Work

Many recent works leverage structured task specifications such as RMs or Linear Temporal Logic (LTL) in deep RL. However, the vast majority of these works do not consider the effects of error or uncertainty in the labelling function. We note a few works that explicitly avoid the assumption of an oracle labelling function. Kuo et al. [32] solve LTL tasks by encoding them with a recurrent policy. Andreas et al. [2] and Oh et al. [43] learn modular subpolicies and termination conditions, avoiding the need for a labelling function, but these approaches are restricted to simple sequential tasks.

Some recent works have considered applying formal languages to deep RL under a noisy evaluation of propositions. Nodari and Cerutti [42] empirically show that current RM algorithms are brittle to noise in the labelling function but do not offer a solution. Tuli et al. [56] teach an agent to follow LTL instructions in partially observable text-based games using the Naive method, but only consider propositions that are observable to the agent. Umili et al. [57] use RMs in visual environments with a method similar to IBU, while Hatanaka et al. [24] update a belief over LTL formulas using IBU, but only update the belief at certain time intervals. While these last three works offer solutions that tackle noise for a specific setting, they do not consider the efficacy of these methods more generally. Hence, we believe our framework and the insights from this paper will provide a useful contribution.

Despite the large corpus of prior work in fully observable settings, RMs and LTL have rarely been considered for general partially observable deep RL problems. We believe this is due to the difficulty of working with an uncertain labelling function when propositions are defined over a partially observed state. The closest work by Toro Icarte et al. [53] applies RMs to POMDPs but only considers propositions defined as a function of observations.

LTL has long been used for specification in motion planning [30], and some works consider an uncertain labelling function or partial observability. However, solutions nearly always depend on a small, tabular state space, while we focus on solutions that scale to infinite state spaces. Ding et al. [13] propose an approach assuming that propositions in a state occur probabilistically in an i.i.d. manner. Ghasemi et al. [19], Verginis et al. [60], Hashimoto et al. [23], and Cai et al. [6] consider a setting where propositions are initially unknown but can be inferred through interactions.

# 8 Conclusion

This work introduces a framework for Reinforcement Learning with Reward Machines where the interpretation or perception of the domain-specific vocabulary is uncertain. We propose a suite of algorithms that allow an RL agent to leverage the structure of the task, as exposed by the Reward Machine, while exploiting prior knowledge through the use of *abstraction models*—preexisting models that noisily ground high-level features into the environment. Through theory and experiments, we show the pitfalls of naively aggregating outputs from a noisy abstraction model, while simultaneously demonstrating how abstraction models that are aware of temporally correlated predictions can mitigate this issue. Ultimately, our techniques successfully leverage task structure to improve sample efficiency while scaling to environments with large state spaces and partial observability.

On the topic of societal impact, our work aims to ameliorate the impact of uncertain interpretations of symbols, which can lead to dangerous outcomes. We note that our proposed methods elucidate the internal decision-making process of RL agents within a formal framework, potentially providing enhanced interpretability. A limitation of TDM, our best-performing method, is that it relies on a task-specific abstraction model (to predict the state in the specific RM). For some real-world tasks, it might not be possible to get enough training data to learn accurate abstraction models (some RM transitions might very rarely be observed), so deploying TDM could lead to poor outcomes.

Our experiments show the promise of leveraging pretrained foundation models (e.g., GPT-4o) as general-purpose abstraction models in a Reward Machine framework. Two of our proposed methods, Naive and IBU, can employ such models directly in many settings where text or image descriptions of the environment are available. A further investigation on the integration of Reward Machines and foundation models is left to future work. Another promising direction is to relax the assumption of access to ground-truth rewards—rewards given by the RM under the ground-truth evaluation of propositions—during training.

## Acknowledgements

We gratefully acknowledge funding from the Natural Sciences and Engineering Research Council of Canada (NSERC), the Canada CIFAR AI Chairs Program, and Microsoft Research. The second-to-last author also acknowledges funding from the National Center for Artificial Intelligence CENIA FB210017 (Basal ANID) and Fondecyt grant 11230762. Resources used in preparing this research were provided, in part, by the Province of Ontario, the Government of Canada through CIFAR, and companies sponsoring the Vector Institute for Artificial Intelligence (`https://vectorinstitute.ai/partnerships/`). Finally, we thank the Schwartz Reisman Institute for Technology and Society for providing a rich multi-disciplinary research environment.

An earlier of this work [33] appeared in the NeurIPS 2022 Deep RL Workshop.

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

# A   Additional Definitions and Theorems

**Theorem 4.1**   A Noisy RM Environment $\langle \mathcal{E}, \mathcal{R}, \mathcal{L}, \mathcal{M} \rangle$ is equivalent to a POMDP over state space $S \times U$ and observation space $O$.

*Proof.*   The POMDP $\mathcal{K} = \langle S', O', A', P', R', \omega', \mu' \rangle$, where $S' = S \times U$, $O' = O$, $A' = A$,

$$P'((s_{t+1}, u_{t+1})|(s_t, u_t), a_t) = P(s_{t+1}|s_t, a_t) \cdot \mathbb{1}[\delta_u(u_t, \mathcal{L}(s_t, a_t, s_{t+1})) = u_{t+1}],$$
$$R'((s_t, u_t), a_t, (s_{t+1}, u_{t+1})) = \delta_r(u_t, \mathcal{L}(s_t, a_t, s_{t+1})),$$
$$\omega'(o_t|(s_t, u_t)) = \omega(o_t|s_t)$$
$$\mu'(s, u) = \mu(s)\mathbb{1}[u = u_1]$$

is equivalent to the Noisy RM Environment $\langle \mathcal{E}, \mathcal{R}, \mathcal{L}, \mathcal{M} \rangle$ in the following sense. For any policy $\pi(a_t|h_t, z_{1:t})$ in the noisy RM environment we can obtain an equivalent policy $\pi'(a_t|h_t) = \pi(a_t|h_t, \mathcal{M}(h_1), \ldots, \mathcal{M}(h_t))$ for $\mathcal{K}$ that achieves the same expected discounted return and vice-versa. To see why the reverse direction is true, we note that $z_{1:t} = \mathcal{M}(h_1), \ldots, \mathcal{M}(h_t)$ is a function of $h_t$ for a fixed abstraction model $\mathcal{M}$. These policies behave identically given the same history $h_t$.   $\square$

**Theorem 4.2** (*Does the choice of $\mathcal{M}$ affect optimal behaviour?*)   Let $\mathcal{P}$ be a Noisy RM Environment $\langle \mathcal{E}, \mathcal{R}, \mathcal{L}, \mathcal{M} \rangle$, and $\mathcal{P}'$ be a Noisy RM Environment $\langle \mathcal{E}, \mathcal{R}, \mathcal{L}, \mathcal{M}' \rangle$. Then $\mathcal{P}$ and $\mathcal{P}'$ are equivalent.

*Proof.*   Consider any policy $\pi(a_t|h_t, z_{1:t})$ for $\mathcal{P}$ and notice that the sequence of outputs from the abstraction model $\mathcal{M}$ is a function of the history $h_t$, i.e. $z_{1:t} = f(h_t)$. Thus, a valid policy in $\mathcal{P}'$ is $\pi'(a_t|h_t, z'_{1:t}) = \pi_*(a_t|h_t, f(h_t))$ (where $z'_{1:t}$ are the outputs from the abstraction model $\mathcal{M}'$ but do not affect the policy $\pi'$). Similarly, a policy for $\mathcal{P}'$ can be used to obtain a policy for $\mathcal{P}$ that is identical. Thus, $\mathcal{P}$ and $\mathcal{P}'$ are equivalent.   $\square$

**Theorem 4.3** (*Does observing $\mathcal{L}$ affect optimal behaviour?*)   Let $\mathcal{P}$ be a Noisy RM Environment $\langle \mathcal{E}, \mathcal{R}, \mathcal{L}, \mathcal{M} \rangle$. Consider a problem $\mathcal{P}'$ that is identical to $\mathcal{P}$ except that the agent at time $t$ additionally observes $\mathcal{L}(s_t, a_t, s_{t+1})$ after taking action $a_t$ in state $s_t$. If $\mathcal{E}$ is an MDP, then $\mathcal{P}$ and $\mathcal{P}'$ are equivalent. If $\mathcal{E}$ is a POMDP, $\mathcal{P}$ and $\mathcal{P}'$ may be non-equivalent.

*Proof.*   Consider that $\mathcal{E}$ is an MDP. At time $t$, denote the sequence of outputs from the labelling function up to time $t$ by $\sigma_{1:t-1}$, where $\sigma_i = \mathcal{L}(s_i, a_i, s_{i+1}) = \mathcal{L}(o_i, a_i, o_{i+1})$ (we assume $s_i = o_i$ since $\mathcal{E}$ is an MDP). Consider a policy $\pi'(a_t|h_t, z_{1:t}, \sigma_{1:t-1})$ for $\mathcal{P}'$ that conditions on all observable information (including the outputs of the labelling function) up to time $t$. However, since $\mathcal{E}$ is fully observable, the labelling function outputs $\sigma_{1:t}$ can be represented as a function of the history $h_t$, i.e. $\sigma_{1:t} = f(h_t)$. Then the policy $\pi(a_t|h_t, z_{1:t}) = \pi'(a_t|h_t, z_{1:t}, f(h_t))$ for $\mathcal{P}$ is identical to $\pi'$. Similarly, given a policy $\pi(a_t|h_t, z_{1:t})$ for $\mathcal{P}$, we can obtain an identical policy $\pi'(a_t|h_t, z_{1:t}, \sigma_{1:t}) = \pi(a_t|h_t, z_{1:t})$ for $\mathcal{P}'$.

Note that if $\mathcal{E}$ is a POMDP, this is not true, since the outputs of the labelling function $\sigma_{1:t-1}$ (which is a function of the history of states and actions) cannot be determined from the history of observations and actions. For an explicit counterexample, consider that $\mathcal{E}$ is a two-state POMDP with states $s^{(0)}$ and $s^{(1)}$ where the initial state is equally like to be $s^{(0)}$ or $s^{(1)}$, and that state persists for the remainder of the episode regardless of the agent's actions. The agent receives a single, uninformative observation $o$ regardless of the state. Let there be two actions $a^{(0)}$ and $a^{(1)}$, where a reward of 1 is received for taking action $a^{(i)}$ in state $s^{(i)}$ (a reward of 0 is received otherwise). Let there be a single proposition $A$ that holds for a transition $(s_t, a_t, s_{t+1})$ if $s_t = s^{(0)}$. For the Noisy RM Environment $\mathcal{P}$ (where the agent cannot observe $\mathcal{L}$), the optimal policy receives no information about the current state and has no better strategy than to randomly guess between $a^{(0)}$ and $a^{(1)}$ at each timestep. In $\mathcal{P}'$ (where the agent can observe $\mathcal{L}$), the agent can deduce the state based on whether $A$ holds or not, enabling a strategy that receives a reward of 1 on each step. There is no identical policy in $\mathcal{P}$ that achieves this, thus the problems are not equivalent.   $\square$

**Proposition A.1**   TDM is consistent.

*Proof.* This is immediate because the belief $\Pr(u_t|h_t)$ is a function of $h_t$. Thus, choose $\mathcal{M}^*(h_t) = \Pr(u_t|h_t)$ and we are done. □

**Lemma A.2** If $\mathcal{E}$ is fully observable, then there exist abstraction models $\mathcal{M}_1 : H \to 2^{\mathcal{AP}}$, $\mathcal{M}_2 : H \to \Delta(2^{\mathcal{AP}})$, and $\mathcal{M}_3 : H \to \Delta U$ such that $\mathcal{M}_1(h_t) = \mathcal{L}(s_{t-1}, a_{t-1}, s_t)$, $\mathcal{M}_2(h_t)[\mathcal{L}(s_{t-1}, a_{t-1}, s_t)] = 1$, and $\mathcal{M}_3(h_t)[u_t] = 1$ for all $h_t \in H$. In other words, $\mathcal{M}_1$, $\mathcal{M}_2$ recover the ground-truth labelling function $\mathcal{L}$ and $\mathcal{M}_3$ recovers the RM state at time $t$ with certainty.

*Proof.* Since $\mathcal{E}$ is an MDP, we assume states $s_t$ and observations $o_t$ are interchangeable. The desired statements immediately result from the fact that the labelling function $\mathcal{L}(s_{t-1}, a_{t-1}, s_t)$ and $u_t$ are deterministic functions of the history $h_t$. □

**Proposition A.3** If $\mathcal{E}$ is fully observable, then Naive and IBU are consistent.

*Proof.* Choose the abstraction models $\mathcal{M}_1$, $\mathcal{M}_2$ and $\mathcal{M}_3$ as in Lemma A.2. Running Naive with $\mathcal{M}_1$ precisely mirrors the procedure for computing RM states $u_t$ with the ground-truth labelling function $\mathcal{L}$, and therefore recovers $u_t$ with certainty.

For IBU with $\mathcal{M}_2$, we perform a simple proof by induction. At any time $t$, we propose that the predicted belief $\tilde{u}_t$ assigns full probability to the ground-truth RM state $u_t$. At time 1, this is true by the initialization of the algorithm. At time $t > 1$, we have

$$
\begin{aligned}
\tilde{u}_t(u) &= \sum_{\sigma \in 2^{\mathcal{AP}}, u' \in U} \mathbb{1}[\delta_u(u', \sigma) = u] \cdot \tilde{u}_{t-1}(u') \cdot \mathcal{M}(h_t)[\sigma] \\
&= \sum_{\sigma \in 2^{\mathcal{AP}}} \mathbb{1}[\delta_u(u_{t-1}, \sigma) = u] \cdot \mathcal{M}(h_t)[\sigma] \\
&= \mathbb{1}[\delta_u(u_{t-1}, L(s_{t-1}, a_{t-1}, s_t)) = u] \\
&= \mathbb{1}[u_t = u]
\end{aligned}
$$

□

**Proposition A.4** Naive is not necessarily consistent in general POMDP environments $\mathcal{E}$.

*Proof.* This result stems from the fact that Naive cannot model an RM state belief of a strictly probabilistic nature.

Consider a POMDP with two states, $s^{(0)}$ and $s^{(1)}$, where the initial state is equally likely to be $s^{(0)}$ or $s^{(1)}$, and that state persists for the remainder of the episode regardless of the agent's actions. The agent receives a single, uninformative observation $o$ regardless of the state. Thus, the agent always observes the same sequence of observations $(o, o, \ldots)$ and the sequence of states is equally likely to be either $(s^{(0)}, s^{(0)}, \ldots)$ or $(s^{(1)}, s^{(1)}, \ldots)$.

Let there be a single proposition $A$ that holds when the agent is currently in state $s^{(0)}$. Let the RM $\mathcal{R}$ have two states: an initial state $u^{(0)}$ that transitions to the state $u^{(1)}$ when the proposition $A$ holds. The ground-truth RM state belief is given by $\Pr(u_t = u^{(1)}|h_t) = 0.5$ for all timesteps $t \geq 2$. This belief cannot be captured by Naive under any abstraction model, thus, Naive is not consistent. □

**Proposition A.5** IBU is not necessarily consistent in general POMDP environments $\mathcal{E}$.

*Proof.* Suppose for a contradiction that IBU is consistent and therefore, produces $\tilde{u}_t = \Pr(u_t|h_t)$ for some $\mathcal{M}^*$.

Consider the same POMDP as in Proposition A.4, with one small change. Now, while in any state $s^{(i)}$ the observation emitted is $o$ with 0.5 probability or $o^{(i)}$ (revealing the state) with 0.5 probability. Consider a history $h_t$ with the observation sequence $o, o^{(1)}$. After seeing the observation $o$, it must be the case that $\tilde{u}_2$ assigns 0.5 probability to both $u^{(0)}$ and $u^{(1)}$, as before. Then after seeing the observation $o^{(1)}$, it becomes certain that the persistent state is $s^{(1)}$, and therefore $\mathcal{PV}(u_t|h_t)$ assigns probability 1 to $u^{(0)}$. However, regardless of how $\mathcal{M}^*(h_t)$ is assigned, it is impossible to achieve

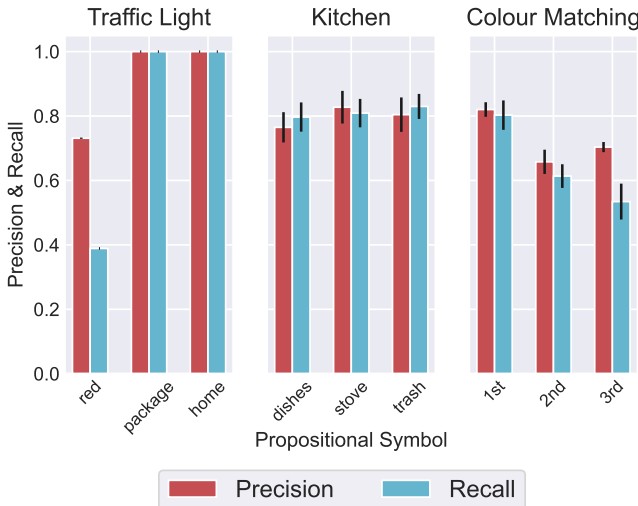

Figure 6: Precision and recall of a classifier trained to predict occurrences of propositions. Key propositions in each domain are uncertain. Values are averaged over 8 training runs, with lines showing standard error.

this belief for $\tilde{u}_3$. This is since $\tilde{u}_2[u^{(1)}] = 0.5$ but there are no RM transitions out of $u^{(1)}$, so $\tilde{u}_3[u^{(1)}] \geq 0.5$. $\qquad\qquad\square$

# B  Experimental Details

## B.1  Gold Mining Experiments

The Gold Mining Problem is described in Example 3.2. We also introduce a small (Markovian) penalty of 0.02 each time the agent selects a movement action to incentivize the agent towards shorter solutions.

**Policy models.** The Oracle baseline learns a Q-value for each (*location*, *RM state*, *action*) combination without memory or function approximation. The Memory only baseline and all the proposed methods are conditioned on six additional binary features corresponding to each square with gold or iron pyrite (i.e. where the agent places non-zero probability of 🪙 ) that indicates whether the agent has previously mined at that location. We use this as a tractable alternative to representing the entire history without neural network encoders.

All approaches except Oracle use a linear decomposition of Q-values to allow for generalization and to reduce the number of estimated Q-values. Naive, IBU, and TDM use the following approximation.

$$Q(location, \tilde{u}, memory_{1:6}, action)$$

$$= \sum_{u \in U} \left[ \tilde{u}(u) \cdot \left[ Q_1(u) + Q_2(location, u, action) + \frac{1}{6} \sum_{i=1}^{6} Q_3(location, u, memory_i, action) \right] \right]$$

The Memory only baseline use the following approximation.

$$Q(location, memory_{1:6}, action) = Q_1(location, action) + \frac{1}{6} \sum_{i=1}^{6} Q_2(location, memory_i, action)$$

**Hyperparameters.** All methods use a learning rate of 0.01, a discount factor of 0.99, and a random action probability of $\epsilon = 0.2$.

**Evaluation details.** All methods are trained for 1M steps, and the policy is evaluated every 10K steps without $\epsilon$-greedy actions. Since the evaluation policy and the environment are deterministic, each evaluation step records the return from a single episode.

## B.2 MiniGrid Experiments

In the Traffic Light domain, the task is described by the RM in Figure 7. There are propositions corresponding to reaching the squares with the package, the home, and for crossing a red light. The light colour cycles between green, yellow, and red, where the duration of the green and red lights are randomly sampled. The yellow light always lasts a single timestep and serves to warn the agent of the red light.

The RM for the Kitchen domain (not shown) has 9 states, encoding all possible subsets of the completed chores plus one additional terminal state that is reached upon leaving the house. The episode is initialized such that each chore needs cleaning with $\frac{2}{3}$ probability, and a chore can be completed by visiting that square. The agent can observe tasks that are completed when inside the kitchen. There is one proposition per chore marking whether that chore is done (regardless of whether the agent can observe this) and a proposition for leaving the house through the foyer. We implement a small cost of $-0.05$ for opening the kitchen door and performing each task, making it so the agent must do so purposefully.

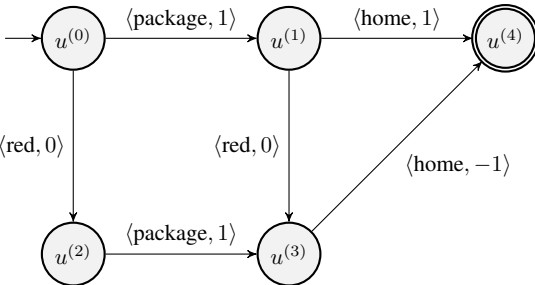

Figure 7: An RM for Traffic Light. The goal is to pick up the package and return home (each stage gives a reward of 1). If the agent crosses a red light, it gets a delayed penalty upon returning home. To simplify formulas on edge transitions, we assume all propositions occur mutually exclusively, and if no transition condition is satisfied, the RM remains in the same state, receiving 0 reward.

**Network architectures:** A policy is composed of an observation encoder followed by a policy network. The observation encoder takes in the current environment observation $o_t$ as well as a belief over RM states (when applicable) and produces an encoding. This encoding is fed to a GRU layer to produce an encoding of the history. The policy network takes the history encoding and feeds it to an actor and critic network, which predicts an action distribution and value estimate, respectively. We use ReLU activations everywhere except within the critic, which use Tanh activations. The observation encoder consists of two 64-unit hidden layers and produces a 64-dimensional encoding. We use a single GRU layer with a 64-dimensional history encoding. The actor and critic both use two 64-dimensional hidden layers. The abstraction models use a separate, but identically structured observation encoder and GRU layer. The history encoding is fed to a single, linear output layer.

## B.3 Colour Matching Experiment

The atomic propositions in the Colour Matching domain are $\{a, b, c, d\}$, where $a$ means the agent is currently in range of the target pillar, and $b$ and $c$ mean the agent is in range of the second and third pillars, respectively. The order of the three pillars is specified at the beginning of the episode via an index vector, where each index corresponds to one of the 18 possible colours.

The RM for this task has 6 states and encodes two pieces of information: whether the target pillar was ever reached, and the last pillar the agent visited. Each time the agent visits an incorrect pillar, a negative reward of -0.2 is issued, but this penalty is not applied if the agent visits the same pillar multiple times in succession. A reward of 1 is given for visiting the target pillar, and then for subsequently going to the portal. We do not show the RM due to the excessive number of edges.

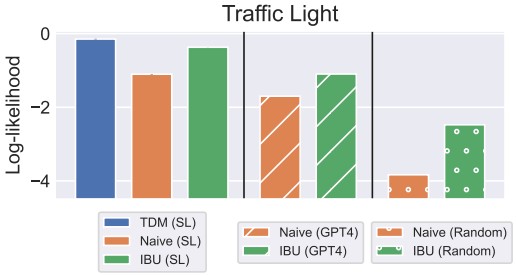
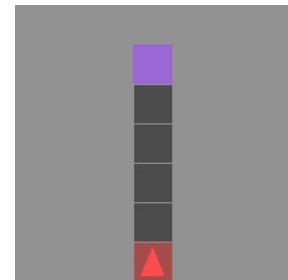

Figure 8: **(Left:)** Accuracy of various inference modules measured by log-likelihood (higher is better) when predicting RM states given trajectories generated by a random policy. (SL:) abstraction model is trained via supervised learning from ground-truth data. (GPT4:) abstraction model is zero-shot GPT-4o. (Random:) abstraction model is a neural network with random weights. **(Right:)** An RGB rendering of the Traffic Light MiniGrid environment used to query GPT-4o. GPT-4o is given the prompt: ``The red triangle is contained in a cell of this grid. What is the color of the cell? Answer in a single word.'' Note that the light colour is only visible when entering the intersection in the forward direction.

Table 2: Hyperparameters for deep RL experiments

| PPO hyperparameters | Traffic Light | Kitchen | Colour Matching |
|---|---|---|---|
| Env. steps per update | 32768 | 32768 | 32000 |
| Number of epochs | 8 | 8 | 8 |
| Minibatch size | 2048 | 2048 | 8000 |
| Discount factor ($\gamma$) | 0.97 | 0.99 | 0.999 |
| Learning rate | $3 \times 10^{-4}$ | $3 \times 10^{-4}$ | $3 \times 10^{-4}$ |
| GAE-$\lambda$ | 0.95 | 0.95 | 0.95 |
| Entropy coefficient | 0.01 | 0.01 | 0.001 |
| Value loss coefficient | 0.5 | 0.5 | 0.5 |
| Gradient Clipping | 0.5 | 0.5 | 5 |
| PPO Clipping ($\varepsilon$) | 0.2 | 0.2 | 0.2 |
| LSTM Backpropagation Steps | 4 | 4 | 4 |
| Abstraction model hyperparameters | | | |
| Env. steps per update | 32768 | 32768 | 32000 |
| Number of epochs | 8 | 8 | 8 |
| Minibatch size | 2048 | 2048 | 8000 |
| Learning rate | $3 \times 10^{-4}$ | $3 \times 10^{-4}$ | $3 \times 10^{-4}$ |
| LSTM Backpropagation Steps | 4 | 4 | 4 |

**Network architectures.** The policy and grounding model networks are similar to those in the previous experiments. The policy uses an observation encoder with two 256-unit hidden layers and a 128-unit output layer. This is fed to a single GRU layer with a hidden size of 128. The actor has one encoding layer with hidden size 128, followed by linear layers to predict a mean and standard deviation for continuous actions. The critic has two 128-unit hidden layers.

The abstraction models for Naive and IBU are MLPs with five 128-unit hidden layers (the environment is nearly fully observable and encoding the history is not necessary for predicting the labelling function). The abstraction model for TDM first predicts evaluations of propositions at each step, and then aggregates the history of these predictions with history of observations, before predicting the RM state belief. We find this approach to work well with a limited dataset as it can also exploit labels corresponding to the evaluations of individual propositions. The network architecture is as follows. An observation encoder first produces a 128-dimensional embedding using three 128-unit hidden layers. This is fed to a decoder that predicts propositional probabilities using one 128-unit hidden layer. The observation embedding and the outputs of this decoder are both fed through 2 GRU layers with hidden size 128, and finally, another decoder uses the memory from the GRU layers to predict the RM state belief.

## B.4 PPO Training Details

We report all hyperparameter settings for the Deep RL experiments in Table 2. Experiments were conducted on a compute cluster. Each run occupied 1 GPU, 16 CPU workers, and 12G of RAM. Runs lasted approximately 18 hours for Colour Matching, and 5 hours for Traffic Light and Kitchen. The Gold Mining experiments took negligible time and were run on a personal computer. The total runtime for these experiments is estimated at 1100 GPU hours. Prior experiments that did not make it into the paper accounted for at least 1000 GPU hours. We used the implementation of PPO found here under an MIT license: `https://github.com/lcswillems/torch-ac`.

## B.5 Agent Behaviours

Table 3 provides a detailed description and explanation of the behaviours we observed for Naive, IBU, and TDM in our experiments in the Gold Mining, Traffic Light, and Kitchen environments.

In the Colour Matching environment, Naive, IBU, and TDM all learn an effective policy in the RL setting, but it is important to note that only TDM is effective at accurately modelling a belief over RM states given trajectories from a random policy (see Figure 5). We believe this is since the RL setting provides leeway for the agent to act in a manner that mitigates the errors of the inference module. In particular, the inference modules are most uncertain when the agent stays near the distance threshold for which a pillar is considered "reached". However, the RL agent can address this by moving closer to the pillar and reducing this uncertainty.

Table 3: Behaviours observed in experiments.

| Method | Description of behaviour | Explanation |
|---|---|---|
| **Gold Mining** | | |
| Naive | The agent typically digs at a nearby cell containing iron pyrite and immediately heads to the depot. | There is a cell that the agent believes yields gold with 0.6 probability but is actually iron pyrite. Under the Naive approach, digging at this location makes the agent believe with certainty that gold has been obtained so the agent heads to the depot. |
| IBU | The agent repeatedly digs at the same locations within the same episode before heading to the depot. | Under IBU, the chance of obtaining gold on each step is assumed to be independent (even when digging at the same square). Hence, the agent repeatedly digs in the same place, even though digging more than once in any location has no utility. |
| TDM | The agent mines at several squares to maximize the chance of obtaining gold before heading to the depot. | Under TDM, we include the assumption that the belief of having obtained gold does not increase on repeated digs at the same square. |
| **Traffic Light** | | |
| Naive | The agent drives through the intersection in reverse (including when the traffic light is red) to pick up the package. | When crossing the intersection in reverse, the agent cannot see the light colour and has to guess. The chance of the light being red tends to be less than 50%, hence the agent predicts that it has not violated the red light. |
| IBU | The agent faces the intersection and waits until the light is green to proceed, successfully solving the task. | IBU is able to capture the uncertainty in whether a red light violation has occurred when crossing the intersection in reverse (as further evidenced by Figure 5). Thus, IBU successfully learns to avoid the dangerous behaviour of driving through the intersection in reverse. |
| TDM | Similar to IBU. | Similar to IBU. |
| **Kitchen** | | |
| Naive | The agent learns the correct behaviour of entering the kitchen and completing the remaining tasks. | Interestingly, the agent's initial belief over RM states is *not* accurate. While outside the kitchen, the agent cannot tell whether each task was initialized in a "done" state or not—it predicts all tasks are *not* done since each starts as "done" with only a $\frac{1}{3}$ probability. This incentivizes the agent to enter the kitchen, allowing it to observe which tasks still need to be completed. |
| IBU | The agent wanders around outside the kitchen and fails to complete the chores. | While outside the kitchen, the agent predicts on each timestep a $\frac{1}{3}$ probability for each chore that it started in a "done" state. As it treats this chance as independent on each timestep, the chance that each chore was done *at some point* up to time $t$ is predicted to be $1 - \left(\frac{2}{3}\right)^t$. By wandering around outside the kitchen for long enough, the agent believes that all chores are completed with close to probability 1, giving it no incentive to enter the kitchen. |
| TDM | The agent learns the correct behaviour of entering the kitchen and completing the remaining tasks. | TDM correctly models the initial belief over RM states reflecting that each chore has an independent $\frac{1}{3}$ chance of being done before the agent has entered the kitchen. This incentivizes the agent to enter the kitchen and complete all chores that may still need to be completed. |

