# OpenReview forum: "Reward Machines for Deep RL in Noisy and Uncertain Environments"
_NeurIPS.cc/2024/Conference — NeurIPS 2024 poster_

### Official Review · Reviewer_pete · 2024-07-06

**Soundness:** 4
**Presentation:** 4
**Contribution:** 3
**Rating:** 8
**Confidence:** 4

**Summary:**

The authors present an extension of the general “Reward Machine” framework to partially observable reinforcement learning environments. In particular, they consider cases where an agent does not have direct access to a labeling function which maps from state transitions to the relevant propositions needed to update the Reward Machine state. They propose a model that learns to directly predict the Reward Machine state from the agent’s trajectory and motivate their approach with both theoretical and experimental results.

**Strengths:**

I do not have any major complaints with this paper. It is well-motivated, well-reasoned, and well-communicated. I particularly appreciate the inclusion of a concrete running example as a way to elucidate the theoretical points made. The empirical experiments are also very thorough, consisting of multiple runs, multiple baselines, and a wide variety of environments.

**Weaknesses:**

If there is a quibble to be made, it is that the paper could benefit from slightly more analysis on the differences between environments. It’s notable that the TDM approach is only one that approximates oracle performance across all 4 environments, but in some environments it is one of many that do so. Is this because those environments are generally less challenging or does some other explanation present itself? I’m slightly surprised that the baselines seem competitive in the color matching environment, in particular, since it is the only non-tabular environment. If the authors have any insight on this issue, it could be nice to include it.

**Questions:**

See above: is there an account for why TDM performance is substantially better on than the baselines on some environments, but not on others?

**Limitations:**

I feel that the authors have adequately addressed the limitations and potential impact of their work.

---

> ### Author Rebuttal · Authors · 2024-08-07
>
> Thank you for your review and for the strong endorsement of our work. We are glad that you recognize its merits across all the major criteria. Please see our response to your feedback and questions below.
>
> > **is there an account for why TDM performance is substantially better on than the baselines on some environments, but not on others?**
>
> This is a great question and something we will certainly spend more time discussing in the paper. As you noted, TDM is the only method that consistently performs well, but Naive and IBU *occasionally* match the performance. We believe there are two reasons this occurs:
>
> 1. In some environments, the simplifying assumptions made by Naive or IBU are reasonable. Thus, the predicted RM state belief may be close to the ground truth RM state belief.
> 2. In other cases, an inaccurate RM state belief can still lead to a reasonable policy.
>
> To elaborate, we can consider each environment.
>
> **Traffic Light**
>
> Recall the main challenge is that the agent can only see the colour of the light when facing it. TDM and IBU both avoid the pitfall of driving backwards through the light because they can model the *chance* the agent may have run a red light. In fact, Figure 4 shows that IBU and TDM predict similarly accurate beliefs under random trajectories — this implies we are in case 1 and indeed IBU learns a reasonable policy on Traffic Light.
>
> Naive doesn’t model the difference between driving through a green light and driving backwards through an unknown light colour (as the probability of the light being red is less than 0.5) and falls into the trap.
>
> **Kitchen**
>
> The correct initial belief should reflect that each chore (independently) has a ⅓ chance of being done (recall some chores may randomly start in the “done” state). This belief should not change until the agent enters the kitchen and observes the true state of all the chores.
>
> IBU fails here by gradually increasing its belief that chores are done over time without entering the kitchen (our response to Q3 from Reviewer sM3w includes a brief explanation of this). This is problematic since IBU conflates two behaviours into the same belief: completing all the remaining chores by entering the kitchen, and only wandering outside the kitchen (which incurs less cost and time to perform than the former, and is preferred by the agent).
>
> Interestingly, Naive also models an incorrect belief but still manages to learn a reasonable policy. Before entering the kitchen, its belief inaccurately reflects that none of the chores are done (as they each have below 0.5 probability). Entering the kitchen allows it to deduce with certainty which chores have been done and which still need to be done. Thus, we are in case 2: Naive models the initial belief incorrectly, but this leads to the same reaction as if it had the correct belief — entering the kitchen.
>
> **Colour Matching**
>
> Recall that the propositions relate to “reaching” certain pillars, which we take to mean the agent is within $d$ distance of the pillar. Unlike in MiniGrid, it is theoretically possible for abstraction models to capture the labelling function perfectly since this environment is an MDP. However, given that our abstraction models were only trained on limited datasets, we noticed that they were often uncertain about propositional values when the distance to a pillar was *approximately* $d$.
>
> This case commonly arises under a random policy, and only TDM captures this uncertainty well. Naive discretely predicts the proposition is either true or false. IBU captures the uncertainty on the first step near the distance threshold $d$, but in MuJoCo, the agent tends to stay near this threshold for several steps. This causes IBU to compound this uncertainty, resulting in a belief that the proposition had occurred with very high probability. These results are reflected in Figure 4.
>
> In the RL setting, the policy has quite a bit of leeway to correct for such errors. We observed that the agent tends to go much closer than distance $d$ to the pillar, resulting in a more certain belief. However, we are not sure if this is intentional or simply a side effect of the agent’s momentum towards the pillar while trying to solve the task *quickly*.
>
> To summarize, only TDM appears to reliably perform well, matching our theoretical understanding since it is the only *consistent* method in POMDPs. Naive and IBU can sometimes perform well, but it depends heavily on the specific task.

---

> > ### Comment · Reviewer_pete · 2024-08-11
> >
> > Thank you for the response. The additional clarifications are much appreciated, and I would encourage the authors to include them in some form in the appendix of the paper.

---

> > > ### Author Response · Authors · 2024-08-12
> > >
> > > These clarifications will be insightful to readers and we will certainly make sure they are included. Thank you again for your positive and constructive review. We greatly appreciate it.

---

### Official Review · Reviewer_qw5f · 2024-07-11

**Soundness:** 2
**Presentation:** 2
**Contribution:** 3
**Rating:** 5
**Confidence:** 2

**Summary:**

This paper focuses on the automatic design of reward machines in reinforcement learning, which holds potential for interpreting instructions, enforcing safety constraints, and more. It is particularly relevant in the real world, especially in the era of large language models (LLMs), where defining reward functions is often challenging. The authors propose to formulate noisy and uncertain environments as Partially Observable Markov Decision Processes (POMDPs), a straightforward approach. Their experiments demonstrate the efficiency of their method. The solution seems broadly applicable across many fields as it doesn't require a ground-truth interpretation of domain-specific vocabulary.

**Strengths:**

The paper introduces a novel way of handling noisy and uncertain environments by modeling them as POMDPs. This perspective is particularly effective in real-world applications where the accurate reward and stationary environment are not always possible.

The automatic design of reward machines is a valuable tool in reinforcement learning, especially in contexts where defining reward functions manually is difficult, such as with LLMs.

**Weaknesses:**

Do the author assume the form of the abstraction model (e.g., classifier or regression model) is known, which might limit its applicability in more complex or less understood environments. In Golden Mining, the modeling of belief as binary might require domain-specific knowledge, which could be a limitation. This simplification might not be suitable for all environments, particularly those requiring more nuanced belief representations. How about the design of the abstraction model in other environments?

Most of the compared methods in the paper are proposed by the authors themselves. This raises questions about whether other existing RL methods could address the noisy and uncertainty environment effectively. A broader comparison with existing methods is necessary to establish the general effectiveness of the proposed approach.


The paper primarily uses Proximal Policy Optimization (PPO) for experiments. It would be beneficial to compare the proposed methods across different RL algorithms to demonstrate consistent improvements and broader applicability.

The relations of RM, noisy RM environment and the evaluating environment is not clear to me. The evaluating environments are just simple POMDP environment.

**Questions:**

What are the constraints of the abstraction model? Or just an arbitrary model. Is its form assumed to be known, such as a classifier or a regression model?

Given that most of the compared methods are proposed in the paper, it is unclear if other RL methods have the potential to solve the discussed noisy reward setting effectively.

How about the interpretability or visualization of the learned model? Could you provide examples of the learned belief or the output of the abstraction model with respect to observations and agent actions? It would be better for the reader to understand.

**Limitations:**

See weakness

---

> ### Author Rebuttal · Authors · 2024-08-07
>
> Thanks for the review. First, we’d like to clarify that our focus is not the “automatic design of Reward Machines” (we assume the RMs are specified by a human). Rather, the work is about whether we can effectively follow task specifications (expressed via RMs) even when the vocabulary cannot be interpreted with certainty. We hope our response further clarifies the work and raises your evaluation.
>
> > **What are the constraints of the abstraction model?**
>
> Our framework allows for abstraction models to be quite general. Abstraction models represent the agent’s uncertain interpretation of high-level salient features, and critically, we allow these features to be arbitrary.
>
> We later focus on three classes of methods (Naive, IBU, TDM) requiring specific types of abstraction models. These abstraction models predict a specific feature (propositional values or RM states) given histories. We agree that requiring specific models is a limitation, but there are two reasons this work is significant nonetheless:
>
> 1. Our work is motivated by the large body of work leveraging RMs or related formal languages in deep RL (see line 32). **The vast majority of these are built on a far stricter requirement that the agent can observe the labelling function — an oracle providing exact propositional evaluations**. Abstraction models significantly relax the labelling function — they can be noisy, they can output arbitrary features, and they depend only on the observable history rather than states.
> 2. **Formal specifications like RMs have already been applied to important real-world problems despite the limitations you mention** (e.g. [1-3]), and they are often more data efficient and easily interpreted compared to end-to-end data-driven approaches.
>
> > **the modeling of belief as binary might require domain-specific knowledge**
>
> You are correct if you are stating that our propositions must be binary. Other formal languages with more expressive vocabularies have also been considered.
> - In Signal Temporal Logic [4], the vocabulary is a continuous-valued signal that can represent values such as distances and velocities.
> - Sun et al [6] consider programs as a specification with a rich set of domain-specific primitives.
> - Tuli et al [5] consider Linear Temporal Logic over open-ended entity relations described in natural language e.g. (“potato”, “is”, “chopped”).
>
> Our work is motivated by the general observation that agents may incorrectly interpret or evaluate their vocabularies, which is relevant to other forms of task specification as well.
>
> > **Most of the compared methods in the paper are proposed by the authors themselves.**
>
> We understand the concern that our experiments mainly use our own methods. **However, we are proposing a novel problem setting and there are few existing approaches that can suitably be applied.** We believe our paper is comprehensive in discussing the relevant literature — for each class of approach, we either include a direct experimental comparison or explain why it is not suitable to our problem.
> - We show our problem can be reduced to a POMDP and compare against Recurrent PPO, a state-of-the-art method for general large POMDPs.
> - There is a broad literature leveraging formal languages in deep RL. These often rely on access to the labelling function and we include the “Oracle” baseline as a performance upper bound.
> - Some applications have used ideas like Naive or IBU to handle noisy inputs (e.g. [5,6]) but these are treated as an implementation trick rather than a general solution for partial observability. We identify the pitfalls of applying such approaches more generally.
> - Many works have considered the noisy detection of propositions in tabular MDPs. Their methods require marginalization over the entire state space which is infeasible in most deep RL environments (e.g. [7]).
>
> > **How about the interpretability or visualization of the learned model?**
>
> We agree that this would be insightful and have already prepared videos of the trained agents. We will ensure these are released in the final version.
>
> > **The paper primarily uses Proximal Policy Optimization (PPO) for experiments.**
>
> We mainly use PPO throughout our experiments as it’s arguably the most popular deep RL algorithm. Nonetheless, the rigorous theory and conceptual examples (as recognized by the other reviewers) supporting our results are independent of the specific policy learning scheme.
>
> > **The relations of RM, noisy RM environment and the evaluating environment is not clear to me**
>
> Unfortunately, we were not completely sure what you meant by this. If you’re asking how our experimental domains depend on RMs (and why they are not just simple MDPs or POMDPs), note that an RM specifies a temporally extended, non-Markovian pattern. For example, in Colour Matching, the agent must touch the correct pillar before entering the portal. A normal RL agent that only considers its position cannot reliably solve this without considering past information (namely, whether the correct pillar was touched yet). The RM state conveniently encodes this salient information, but the RM state cannot be computed with certainty in our noisy problem setting.
>
> [1] Fainekos et al. "Temporal logic motion planning for dynamic robots." Automatica, 2009.
>
> [2] Doherty et al. "A temporal logic-based planning and execution monitoring framework for unmanned aircraft systems." Autonomous Agents and Multi-Agent Systems 2009.
>
> [3] Camacho et al. "Reward machines for vision-based robotic manipulation." ICRA, 2021.
>
> [4] Aksaray et al. "Q-learning for robust satisfaction of signal temporal logic specifications." IEEE Conference on Decision and Control, 2016.
>
> [5] Tuli et al. "Learning to follow instructions in text-based games." NeurIPS, 2022.
>
> [6] Umili et al. "Visual reward machines." Neural-Symbolic Learning and Reasoning, 2022.
>
> [7] Ghasemi et al. "Task-oriented active perception and planning in environments with partially known semantics." ICML, 2020.

---

> > ### Comment · Area_Chair_ktLD · 2024-08-11
> >
> > Dear Reviewer qw5f, does the author's response change your assessment or do you still have concerns? It would be helpful to raise them now while we can still ask the authors for more clarification.

---

> > ### Comment · Reviewer_qw5f · 2024-08-11
> >
> > Thank you for your response. Most of my concerns have been addressed. I will raise my score.

---

> > > ### Author Response · Authors · 2024-08-12
> > >
> > > We're happy to hear that we've managed to alleviate most of your concerns. Thanks again for the review, and if there are any outstanding concerns that we can address, please let us know.

---

### Official Review · Reviewer_ed91 · 2024-07-12

**Soundness:** 2
**Presentation:** 4
**Contribution:** 3
**Rating:** 7
**Confidence:** 3

**Summary:**

The paper proposes the use of Reward Machines (RMs) in deep reinforcement learning (RL) for noisy and uncertain environments, characterizing these settings as Partially Observable Markov Decision Processes (POMDPs). The contributions include:
- Proposing framework for using RMs in deep RL in partially observable environments.
- Theoretical analysis identifying pitfalls in naive approaches.
- Experimental results demonstrating improved performance under noisy conditions.
- Discussing limitations and proposing future work for general-purpose models and relaxing ground truth reward assumptions.

**Strengths:**

Originality:
- The combination of RMs with deep RL algorithms to handle noisy and uncertain environments is innovative.
- The paper provides rigorous mathematical definitions and theoretical insights into the limitations of naive approaches and Independent Belief Updates (IBU).

Quality:
- The submission is technically sound with well-supported claims through both theoretical analysis and experimental results.
- The experiments are comprehensive, spanning toy environments to more realistic tasks, showing the scalability and applicability of the methods.

Clarity:
- The paper is generally well-written and organized.
- The explanation of theoretical concepts using the Gold Mining Problem is clear and effective.

Significance:
- The results demonstrate significant improvements in RL performance and sample efficiency in noisy environments.
- The experimental evaluation shows the potential for real-world applications.

**Weaknesses:**

- It is unclear how well the methods perform without ground-truth RMs, raising questions about additional effort required for new environments.
- The discussion on foundational models is mentioned early on but is missing from the experiments, discussion, and conclusion, leaving the practical setting for these methods unclear.
- Line 32 mentions numerous references without distinguishing their importance. This should be extended to provide (even if just brief one-sentence) summaries of the mentioned works.

**Questions:**

The proposed technique makes build a model for a reward machine either from data or via pretrained foundational labeling functions. making such GT data available to a policy would likely increase training performance: By providing useful features of the environment instead of raw observations, the difficulty of the task the neural network has to solve is reduced. But this is already given by the No-Free-Lunch-Theorem.

The evaluations show the policies trained with the newly proposed methods to perform better, but to what extend could this be explained by 'leaking' information of the ground thought RM to the policy during training? Can we actually make predictions on whether this is an improvement when no GT RM data is available (as such leaks can no longer occur)?

**Limitations:**

The paper mentions addresses some limitations, such as the need for ground-truth rewards during training. However, could be extended by further discussion on the practical challenges of implementing these methods in real-world scenarios.

---

> ### Author Rebuttal · Authors · 2024-08-07
>
> Thank you for your positive evaluation and for recognizing that this work addresses an innovative problem, introduces rigorous definitions and insights, and presents well-supported claims in a clear and organized manner. We address your main questions and concerns below.
>
> > **… to what extent could [evaluations] be explained by 'leaking' information of the ground truth RM to the policy during training? Can we actually make predictions on whether this is an improvement when no GT RM data is available? …**
>
> Indeed, an abstraction model serves to identify salient features of the problem such as the ground-truth propositional evaluations or RM states. You are correct that exposing such features can allow for significant performance benefits. This is one of the merits of RMs (and similar formal specifications) — they leverage this prior information in a systematic way to yield greater sample efficiency and interpretability. In fact, a number of real-world applications already depend on these types of methods, e.g. [1-3].
>
> The last part of your question is also important: what happens when abstraction models are unable to “leak” these relevant features? Firstly, an abstraction model that provides no signal at all is entirely noise (i.e. the model outputs are independent of the features, such as a neural network with random weights). In such cases, we hypothesize that exploiting the reward function structure is nearly impossible since we have no information regarding the semantics of the propositional symbols.
>
> This paper relaxes a strong assumption inherent in many RL works leveraging formal languages for reward function specification, including works on Reward Machines, and various temporal logics (e.g. Metric Interval Temporal Logic [4], Linear Temporal Logic [5]). These approaches are often predicated on a "perfect" labelling function, an oracle that returns ground-truth features for any environment transition. In contrast, abstraction models are significantly easier to obtain — they can be noisy, can model arbitrary features (not just propositions), and depend only on the observable history (not states). This allows us to better handle practical concerns such as ambiguity in the intended interpretation of propositions, noisy sensors, and partial observability. As you noted, foundation models fit into our framework too and bringing to bear such models presents an exciting direction for this field. For these reasons, we consider our work a step towards making RMs more widely applicable in the real world.
>
> We understand your concern that we may “leak” too much ground-truth information through the abstraction models in our experiments. However, Figure 5 in the Appendix shows that these abstraction models are in fact quite noisy — they have poor precision and recall when predicting the most important propositions (e.g. running a red light in Traffic Light, or the completion of chores in Kitchen) under a random policy. Also note that in our partially observable domains, the abstraction models cannot capture the ground-truth labelling function or RM state even with infinite data. We fundamentally restrict abstraction models to depend only on observable histories, while ground-truth propositional values and RM states are functions of state.
>
> > **The discussion on foundational models is mentioned early on but is missing from the experiments, discussion, and conclusion.**
>
> An RM framework that incorporates foundation models is an important and exciting direction. We are happy to include a further discussion on this topic.
>
> We’ve conducted additional experiments using vision-language models in the Traffic Light MiniGrid domain, showing that GPT4o can serve as an effective zero-shot abstraction model (see global response). We consider the task of predicting RM state beliefs from a dataset of randomly generated trajectories, like in the original Figure 4, and we directly prompt the VLM to list the coloured grid squares it can see from the agent’s image observation. We will discuss these capabilities in a subsection of “Experiments”.
>
> We will also discuss the limitations of current foundation models. Namely, we find that only GPT4o effectively understands MiniGrid observations, while smaller models (GPT4o-mini, CLIP) fail. There is also no easy way to implement TDM, the most robust type of abstraction model, using current VLMs.
>
> > **Line 32 mentions numerous references without distinguishing their importance**
>
> We originally included the list of references on line 32 to convey the depth of literature on formal specifications (particularly those based on LTL or automata) in deep RL. This helps establish that we are working on an important problem of broad interest to the research community. Nonetheless, your point is well taken and we are happy to provide short contrastive descriptions of these works in an extended related works section.
>
> [1] Fainekos et al. "Temporal logic motion planning for dynamic robots." Automatica, 2009.
>
> [2] Doherty et al. "A temporal logic-based planning and execution monitoring framework for unmanned aircraft systems." Autonomous Agents and Multi-Agent Systems, 2009.
>
> [3] Camacho et al. "Reward machines for vision-based robotic manipulation." ICRA, 2021.
>
> [4] Xu & Topcu. "Transfer of temporal logic formulas in reinforcement learning." IJCAI, 2019.
>
> [5] Vaezipoor et al. "Ltl2action: Generalizing ltl instructions for multi-task rl." ICML, 2021.

---

> > ### Comment · Reviewer_ed91 · 2024-08-10
> >
> > Thank you for the additional clarifications and the thoughtful response to my concerns. I appreciate the detailed explanations regarding the potential "leakage" of ground-truth information and the role of abstraction models in your experiments.
> >
> > I also commend your efforts to incorporate further discussion on foundational models, including the additional experiments with vision-language models. It's clear that you’ve put significant thought into expanding the practical relevance of your work, which is commendable.
> >
> > Given the thoughtful revisions and effective responses to key concerns, the paper's quality has been notably improved. Theoretical solidity, added clarity, and broader contextualization have strengthened the work.
> >
> > I have therefore upgraded my rating from "Weak Accept" to "Accept."

---

> > > ### Author Response · Authors · 2024-08-12
> > >
> > > Thanks again for your constructive comments. We completely agree that the broader contextualization with respect to foundation models and the clarification of the role of abstraction models are an improvement. We will ensure these key points are included in the final paper.

---

### Official Review · Reviewer_sM3w · 2024-07-12

**Soundness:** 2
**Presentation:** 2
**Contribution:** 2
**Rating:** 5
**Confidence:** 3

**Summary:**

This paper investigates the use of Reward Machines in Deep Reinforcement Learning (RL) for handling noisy and uncertain environments. It frames the problem as a Partially Observable Markov Decision Process (POMDP) and proposes a set of RL algorithms leverage the task structure under uncertain interpretation  of domain-specific vocabulary. The theoretical analysis reveals the limitations of naive approaches, while experimental results demonstrate that the proposed algorithms successfully leverage task structure to improve performance under noisy interpretations. The findings provide a general framework for exploiting Reward Machines in partially observable environments.

**Strengths:**

1. The discussion of reward machine under uncertain interpretation of the domain-specific vocabulary is somewhat novel.

2. The paper discusses three possible abstraction models that predicts deterministic proposition, stochastic proposition, and stochastic RM state respectively.

3. The paper provides theoretical analysis on the consistency of these three different abstraction models.

**Weaknesses:**

1. I think this paper could be improved by clearer presentation. The formulation of Noisy Reward Machine Environment is confusing. Especially, the abstraction model is included as part of the environment or part of the problem setting, however, the proposed methods considers different types of abstraction models. Should the abstraction model be the method or the problem?

2. The paper claims as the first to consider reward machines under environment uncertainty. However, its discussion and theoretical analysis does not go much beyond the POMDP and brief state updating in the existing literature. For example, its result can be seamlessly handled by including RM state U as part of the POMDP state.

3. Some writing suggestions:
- In the introduction, the authors used term "new RL framework", but up to Section 5, there is no RL. The three methods being discussed are abstraction models. The authors didn't talk much about how the abstraction models can be combined with RL.
- Make it clear about the problem setting. In particular, is the abstraction of model given as a part of the problem?
- The author used term "optimal behavior", "optimal brief". These term should be clearly defined. What are the optimal behavior? Optimal assuming accessing to label function L? What is optimal state brief?
- Line 222 "Given an abstraction model of the form M : H → ∆U predict M(h_t) directly." what does this mean?

**Questions:**

1. Could authors provide a clear definition of the problem (if abstraction of model is part of the problem) and also provide a definition of optimality under the definition of the problem?

2. In the experiments, abstractions models are represented by neural networks. Where do the ground truth labels come from? Especially, Naive, IBU, and TDM have different prediction objective. So they use different training data and labels?

3. In three of the domains, IBU is even worse than naive that do not consider stochasticity? Could authors explains why this is happening?

**Limitations:**

TDM, the best-performing method, relies on a task-specific abstraction model (to predict the state in the specific RM). For some real-world tasks, it might not be possible to get enough training data to learn accurate abstraction models.

---

> ### Author Rebuttal · Authors · 2024-08-07
>
> Thank you for your constructive review. We take seriously the issues you raised regarding clarity and will revise the manuscript accordingly.
>
> > **Should the abstraction model be the method or the problem?**
>
> The abstraction model is part of the problem. It captures the agent’s uncertain prior over how high-level features are grounded in the environment. The fact that this uncertainty arises and is not easily resolved in many real-world domains motivates our work. Thus, we include this uncertainty as an element of the problem rather than something within our freedom to affect. In terms of the Gold Mining example, we argue that the agent’s uncertainty of where gold can be found should be stated in the problem, not the solution. Thanks for raising this point of potential confusion — we will include this justification when introducing our framework.
>
> Regarding your concern that the proposed methods require different types of abstraction models, Theorem 4.2 establishes that, under any choice of abstraction model, the problems are equivalent (i.e, per the definition in the paper, that for each pair of problems, there is a bijection between policies of equal value.)
>
> > **Could authors provide … a definition of optimality …**
>
> Thanks for pointing out that *optimal behaviour* is potentially ambiguous. In this work, it refers to the behaviour that maximizes expected discounted return in the Noisy RM environment. Rewards are defined by the RM interpreted under the labelling function, $\mathcal{L}$, and we do not assume the agent can query $\mathcal{L}$ when executing.
>
> The *optimal RM state belief* is defined as the distribution $P(u_t | h_t)$ (line 160). It is closely related to a POMDP belief state distribution, $P(s_t | h_t)$. As such, the optimal RM state belief can be viewed as inferring the ground-truth RM state from histories (which depends on $\mathcal{L}$), marginalized over all possible state trajectories, while Naive, IBU, and TDM approximate this optimal belief.
>
> We'll clarify both of these concepts in the paper.
>
> > **Where do the ground truth labels come from? Especially, Naive, IBU, and TDM …**
>
> We obtained offline datasets to train the abstraction models as follows. We generated full episodes from a random policy and annotated each timestep with the ground-truth propositional evaluations $\sigma_t \in 2^\mathcal{AP}$ and RM state $u_t \in U$ obtained via a manually constructed labelling function.
>
> Naive and IBU were trained to predict $\sigma_t$ given $h_t$, while TDM was trained to predict $u_t$ given $h_t$. The abstraction models were trained from the same set of trajectories, ensuring a fair comparison between Naive, IBU, and TDM (even though the target labels were different).
>
> > **In three of the domains, IBU is even worse than naive … Could authors explains why this is happening?**
>
> Naive and IBU are flawed in different ways when predicting a belief over RM states — it should not be concluded that one is generally better than the other. The key flaw with IBU is that when a proposition is uncertain, IBU ignores the dependence of that proposition across timesteps.
>
> Example 5.2 illustrates this. When the agent mines repeatedly at the same state, the outcomes are linked — either there is gold at every time, or at none of the times. By assuming independence, IBU incorrectly assigns non-zero probability to cases that cannot occur, such as the first “mine” action yielding gold but not the second one. Unfortunately, this can mislead the agent into undesirable behaviours such as mining at the same location over and over, to maximize the perceived probability of obtaining gold.
>
> In the Kitchen task, the agent initially cannot observe the state of the kitchen. Consider an episode where the agent never enters the kitchen. Then at each step $t$, the agent (correctly) believes there is a small chance the dishes are already clean. However, IBU ignores the dependence between these events — in reality, the dishes are either clean at every step or at none of the steps. Similar to Gold Mining, IBU will erroneously reflect that all chores are complete with probability approaching 1, even when the agent never enters the kitchen.
>
> > **Some writing suggestions**
>
> Very good suggestions. Thank you.
> - We will include an algorithm box describing how we combine RL with abstraction models (see the global response pdf).
> - To your question, in the description of TDM (line 222), the output $\mathcal{M}(h_t)$ of the abstraction model has the required form (a distribution over RM states) to be used directly as the output of the inference module. We will clarify this and also add a description of an instantiation of TDM where an RNN directly predicts a distribution over RM states given the history.
>
> > **The paper claims as the first to consider reward machines under environment uncertainty ... [the] result can be seamlessly handled by including RM state U as part of the POMDP state**
>
> To clarify, we provide the first scalable deep RL framework for RMs (and related formal languages) under noisy determination of propositions. We respectfully disagree that our results follow seamlessly from viewing the RM state as part of the POMDP state. Characterizing the problem as a POMDP relates it to established concepts and supports theoretical analysis. However, this insight alone does not yield an effective RL solution because it neglects to exploit the rich reward function structure provided by the RM. This is reflected in the poor performance of the generic “Memory only” baseline (Recurrent PPO). Our proposed methods are so effective because they exploit the structure of our specific POMDP formulation.
>
> The related work section elaborates upon the novelty of our framework in comparison to previous works.
>
> #### **Limitations**
> Agreed.  These were explicitly acknowledged as limitations in the paper.

---

> > ### Comment · Reviewer_sM3w · 2024-08-09
> > **Response to abstraction model problem or method**
> >
> > Thanks for the clarification. As the author pointed out, the abstraction model belongs to the problem rather than the method. So Section 5 is about three methods proposed for three different problems settings, but in many discussion, the authors seem to be drawing comparison between them. Also, in section 6, the three methods are directly compared, although they belongs to different problems. Also, does the Memory-only method uses the feature from the abstraction model? If it do not assume access to RM structure, like RM state space, it should at least directly takes abstraction model output as input to draw fair comparison.
> >
> > Also, what conclusion do we want to draw from these comparison? The authors listed 3 questions in line 239, but I don't see any of these being addressed through the experiments.
> >
> > I really think this paper could benefit from a clearer statement of the problem, and rephrase its discussion and analysis.
> >
> > Also, it would be better to mention some real-world cases when different forms abstraction exists.

---

> > ### Comment · Reviewer_sM3w · 2024-08-09
> > **Optimality**
> >
> > I am still confused by the optimal behavior. Imagine a environment of just one step. Either move left or move right. The agent does not know which one will give a reward of +1. The best it can do is random guess. Is random guess the optimal behavior?
> >
> > I also couldn't understand Theorem 4.2. How could the choice of $\mathcal{M}$ not affecting optimal behavior? An arbitrarily bad \mathcal{M} provides zero information, while an ideal $\mathcal{M}$ that deterministically predicts RM state u recovers the MDP.
> >
> > In terms of optimal belief, line 160 is just a notation. I still don't understand what make a distribution $P(u_t \mid h_t)$ optimal? Or are you saying an optimal belief state is the distribution that deterministically predict the ground truth RM state $u$?

---

> > > ### Author Response · Authors · 2024-08-10
> > > **Re: sM3w Optimality & Response to abstraction model problem or method (Part 1)**
> > >
> > > Thanks for following up with further questions.  We appreciate the considerable effort this takes and we are committed to resolving your outstanding issues to ensure the paper is clear.
> > >
> > > We elected to address your two official comments in one (split) response.
> > >
> > > We preface our answers to your questions with four critical points that, based on your questions, we conjecture may assist in resolving your questions.
> > >
> > > [1] **The abstraction model does not affect what optimal behaviour is.** This is a consequence of [2] and [3] below. However, it can impact the difficulty of *identifying* and *learning* the optimal behaviour.
> > >
> > > [2] **The abstraction model $\mathcal{M}$ does not affect the set of behaviours the agent can perform.** Recall that we are interested in policies of the form $\pi(a_t | h_t, z_{1:t})$ (line 104), where $z_i = \mathcal{M}(h_i)$ and $h_t$ is the history of observations and actions up to time $t$. This is actually no more expressive than policies of the form $\pi(a_t | h_t)$ (i.e. Recurrent PPO) since $z_{1:t}$ is a deterministic function of $h_t$. Thus, the set of behaviours under consideration in our problem setting is precisely the set of history-based policies (i.e. policies $\pi : H \to \Delta A$), and is independent of $\mathcal{M}$.
> > >
> > > [3] **The objective (the expected discounted return) is independent of $\mathcal{M}$.** Recall that rewards are given by the RM $\mathcal{R}$ interpreted under the (hidden) labelling function $\mathcal{L}$. Thus, for any behaviour, we can determine its expected discounted return independent of $\mathcal{M}$.
> > >
> > > [4] **For most problems, there exists no abstraction model that can recover the propositions or RM state with certainty.** This is because our framework allows the environment to be *partially observable*, with propositions and RM states depending on the (hidden) POMDP state. However, abstraction models are functions of the observable history $h_t$.
> > >
> > > ### Re: sM3w Optimality
> > >
> > > > Imagine a environment of just one step. Either move left or move right. The agent does not know which one will give a reward of +1. The best it can do is random guess. Is random guess the optimal behavior?
> > >
> > > Why does the agent not know the optimal action?
> > >
> > > If it’s because the optimal action depends on the initial state (which is randomized), and the agent can’t distinguish the initial state given the initial observation, then yes, guessing is an optimal behaviour. There is no better policy given the observation history.
> > >
> > > However, if in every episode the same action (say, “left”) is always better, and the agent does not initially know this (as is typical in RL), the optimal behaviour is to always go “left”. There exists a policy that does this and it maximizes expected return.
> > >
> > > > How could the choice of $\mathcal{M}$ not affecting optimal behavior?
> > >
> > > Please see [1,2,3]. To be clear, given any policy $\pi(a_t | h_t, z_{1:t})$ for an arbitrary abstraction model $\mathcal{M}$, there is a corresponding policy $\pi’(a_t | h_t)$ that encodes the same behaviour as $\pi$. Intuitively, this is because the abstraction model outputs $z_{1:t}$ are a function of $h_t$ and therefore the equivalent computation can be performed directly via the policy $\pi’$. This guarantees that the maximum expected discounted return is the same for any abstraction model $\mathcal{M}$, as stated in Theorem 4.2. What Theorem 4.2 does *not* state is how easy or hard it is to learn this optimal policy.
> > >
> > > Much of our analysis in these points is used to prove Theorems 4.1-4.3 in the paper, but we would be happy to make it more visible in the main paper if the reviewer thinks it would clarify the setting.
> > >
> > > > an ideal $\mathcal{M}$ that deterministically predicts RM state u recovers the MDP
> > >
> > > Generally, abstraction models are fundamentally unable to recover propositional values or RM states with certainty (per [4]). However, in the special case of a fully observable environment, we agree that such an abstraction model exists (Lemma A.2 in the Appendix).

---

> > > > ### Author Response · Authors · 2024-08-10
> > > > **Re: sM3w Optimality & Response to abstraction model problem or method (Part 2)**
> > > >
> > > > > I still don't understand what make a distribution $P(u_t | h_t)$ optimal? Or are you saying an optimal belief state is the distribution that deterministically predict the ground truth RM state?
> > > >
> > > > Related to [4], you cannot always deterministically predict the true RM state from the history in a POMDP environment — multiple different rollouts of environment states $s_1, …, s_t$ (yielding different RM states) can result in the same history $h_t = (o_1,...,o_t)$. The optimal RM state belief simply marginalizes over all possible rollouts of states given the history to infer the RM state distribution, i.e. in finite-state POMDPs
> > > >
> > > > $$P(u_t | h_t ) = \sum_{s_{1:t} \in S^t}  P(u_t | s_{1:t}) P(s_{1:t} | h_t)$$
> > > >
> > > >
> > > > Here, $P(u_t | s_{1:t})$ is deterministic — it is the RM state given the sequence of states (under the true labelling function), while $P(s_{1:t} | h_t)$ depends on the POMDP transition function and observation probabilities. As you can imagine, exact computation of this belief is typically infeasible and we must instead resort to approximate inference of the RM state.
> > > >
> > > > If we restrict ourselves to fully observable environments, then indeed the ground-truth RM state is deterministic given the history. In this case, the optimal belief deterministically predicts the ground-truth RM state $u_t$ as you said.
> > > >
> > > > ### Re: sM3w Response to abstraction model problem or method
> > > >
> > > > > in section 6, the three methods are directly compared, although they belongs to different problems
> > > >
> > > > Yes, in comparing Naive, IBU, and TDM, we are considering three Noisy RM Environments where (only) the abstraction models differ. This is the disadvantage of including $\mathcal{M}$ as part of the problem setting, but per our original response, it is also important to frame the agent’s uncertainty over the vocabulary as part of the problem. [2] and [3] ensure us that this comparison is still meaningful — the space of possible behaviours and the objective remain the same in all three cases.
> > > >
> > > > > does the Memory-only method uses the feature from the abstraction model?
> > > >
> > > > Memory-only does not observe outputs from the abstraction model since it represents solving the task end to end without knowledge of the RM or propositions (per Theorem 4.1). We’ve run a baseline on MiniGrid that’s more along the lines of what you’re suggesting: Recurrent PPO that also observes (noisily) predicted propositions. On Traffic Light, it makes no improvement over “Memory only”. On Kitchen, it marginally outperforms Naive but is worse than TDM. This agrees with our claims that under noisy abstraction models, Naive and IBU can be detrimental, but TDM consistently performs well.
> > > >
> > > > > what conclusion do we want to draw from these comparison? The authors listed 3 questions in line 239, but I don't see any of these being addressed through the experiments.
> > > >
> > > > We conclude the following regarding the three experimental questions:
> > > > - Naive and IBU are sometimes effective for RL. TDM reliably outperforms Recurrent PPO (Fig 3), demonstrating the importance of leveraging RM structure.
> > > > - Naive and IBU can be brittle when abstraction models are noisy due to partial observability, limited training data, or generalization error. This can lead to unintended or unsafe outcomes when training with RL.
> > > > - TDM models the RM state belief more accurately than Naive, IBU (Fig 4) when abstraction models are trained from equivalent data (per our original rebuttal).
> > > >
> > > > These are referenced in our results but we can state them more directly.
> > > >
> > > > > Also, it would be better to mention some real-world cases when different forms abstraction exists.
> > > >
> > > > Our experiments show that a dataset of trajectories labelled with propositional values can be used to train abstraction models for TDM, Naive, and IBU, with TDM being the most effective.
> > > >
> > > > In some cases, different modelling assumptions can also be used to obtain different abstraction models for the same domain. For a (non-real-world) example, in Gold Mining, Naive, IBU, and TDM all leverage the same belief about where gold can be found but make different assumptions to predict an RM state belief (Sections 5.1-5.3).

---

> > > > > ### Comment · Area_Chair_ktLD · 2024-08-11
> > > > >
> > > > > Thanks Reviewer sM3w for actively engaging with the authors to discuss your concerns -- I appreciate it and I'm sure the authors do too! Please let me and the authors know if you still have concerns following the discussion.

---

> > > > > > ### Comment · Reviewer_sM3w · 2024-08-12
> > > > > > **Response**
> > > > > >
> > > > > > Thanks for the clarification. Now I am more clear about optimality, but I still believe those terms are not precisely defined in the paper. For example, line 160 is not a definition, just a notation. The authors need to include some explanations like the rebuttal shows. I have raised my evaluation accordingly.

---

> > > > > > > ### Author Response · Authors · 2024-08-12
> > > > > > > **Thanks**
> > > > > > >
> > > > > > > We will be sure to include these explanations in the paper and we thank you once again for the productive discussion. Please let us know if you have any final questions or concerns that we can address.

---

### Author Rebuttal · Authors · 2024-08-07

Thank you to the reviewers for their time and for their detailed and informative reviews. Reviewers found “the combination of RMs with deep RL algorithms to handle noisy and uncertain environments [to be] innovative” (ed91), noting that the work was “broadly applicable across many fields as it doesn't require a ground-truth interpretation of domain-specific vocabulary” (qw5f). Reviewers praised the inclusion of “rigorous mathematical definitions and theoretical insights” (ed91), and recognized our experimentation as “very thorough” (pete).

Reviewers raised a number of good points. In the individual reviews that follow, we have addressed the major questions raised by reviewers, together with additional comments. We are confident that we can address all feedback in the final paper. In two instances, we were uncertain whether we had correctly interpreted reviewer comments. If we have not adequately addressed a question in the review, please raise this during the discussion period. We appreciate reviewers’ engagement with the material and are keen to answer reviewer questions.

Some of the main points are summarized below.

1. [**A subset of reviewers suggested improvements to the presentation.**]

Reviewer sM3w identified details relating to our framework that were unclear. We will make the following changes to improve readability for the broader NeurIPS audience:

- We’ll include an architectural diagram (global pdf, Fig. 1) to clarify the elements of our framework including which elements are available during training and execution.
- We’ll include an algorithm box (global pdf, Fig. 2) that provides a more precise description of how we incorporate abstraction models into RL.

Reviewers sM3w, qw5f, and pete suggested further elaboration of the experimental results. We will provide a more detailed explanation of the experimental outcomes in line with our answers to reviewer questions. We have also prepared videos of the trained agents to be released with the final version.

2. [**Further discussion of foundation models.**]

Reviewer ed91 requested further discussion on the possibility of foundation models as abstraction models. We agree that this adds to the paper, and have conducted some preliminary experiments using GPT-4o as a zero-shot abstraction model.

We considered the Traffic Light MiniGrid domain, where we rendered RGB images of the environment and prompted GPT-4o to detect if the agent was standing on a coloured grid cell (representing a propositional occurrence). We found that GPT-4o could determine propositional values to predict accurate RM state beliefs with our methods, achieving performance close to a model trained on ground-truth data, and significantly outperforming a random abstraction model (global pdf, Fig. 3).

We will further discuss the limitations of current foundation models. In particular, these models provide no easy way to implement TDM (which had the most robust performance in our experiments), and only the largest available vision-language models were able to interpret the MiniGrid images correctly.

3. [**Dependence on domain knowledge and/or ground-truth features.**]

Reviewer qw5f mentioned that our noisy RM framework requires domain knowledge to specify propositions and abstraction models, and Reviewer ed91 noted that abstraction models can potentially leak relevant features to the learning algorithm, particularly when trained on ground-truth data.

We acknowledge that these are weaknesses, but it’s important to understand why our work is significant nonetheless. RMs and related formal languages are so effective for specifying reward functions because they expose a task’s logical and temporal structure over an abstract vocabulary. There is a rich literature that leverages these specifications to synthesize controllers with provable guarantees when an environment model is available, but they can also significantly improve the performance and interpretability of deep RL algorithms in the absence of such a model.

Up until now, works leveraging these specifications largely depended on the availability of “perfect” labelling functions that identify ground-truth features for any environment transition. Our work relaxes this strong assumption by instead relying on abstraction models, which can be noisy. As shown above, foundation models are a promising way of implementing abstraction models in new environments, and they will likely be noisy. We consider our work an important step towards making RMs more widely applicable in the real world.

---

### Decision · Program_Chairs · 2024-09-25

**Decision:**

Accept (poster)

**Comment:**

This paper investigates the use of Reward Machines in Deep Reinforcement Learning (RL) for handling noisy and uncertain environments. It frames the problem as a Partially Observable Markov Decision Process (POMDP) and proposes a set of RL algorithms leverage the task structure under uncertain interpretation of domain-specific vocabulary. The theoretical analysis reveals the limitations of naive approaches, while experimental results demonstrate that the proposed algorithms successfully leverage task structure to improve performance under noisy interpretations. The findings provide a general framework for exploiting Reward Machines in partially observable environments.

Stengths: the reward machine formalsim holds potential for interpreting instructions, enforcing safety constraints, and more. This paper extends its use to the partially-observable setting which is significant for real-world use of RL. Reviewers appreciated the theoretical and empirical analysis provided. The paper is clearly written.

Weaknesses: the method requires a ground-truth reward function and this can be seen as an immediate limitation. There was a concern about fairness of empirical comparisons but that concern has been addressed with discussion.

The paper was extensively discussed and all major concerns were addressed by the author. The authors should make all clarifications requested by the reviewers.